# UniMTS: Unified Pre-training for Motion Time Series

**Xiyuan Zhang**
UC San Diego
xiyuanzh@ucsd.edu

**Diyan Teng**
Qualcomm
diyateng@qti.qualcomm.com

**Ranak Roy Chowdhury**
UC San Diego
rrchowdh@ucsd.edu

**Shuheng Li**
UC San Diego
shl060@ucsd.edu

**Dezhi Hong**[*]
Amazon
hondezhi@amazon.com

**Rajesh K. Gupta**
UC San Diego
rgupta@ucsd.edu

**Jingbo Shang**
UC San Diego
jshang@ucsd.edu

## Abstract

Motion time series collected from low-power, always-on mobile and wearable devices such as smartphones and smartwatches offer significant insights into human behavioral patterns, with wide applications in healthcare, automation, IoT, and AR/XR. However, given security and privacy concerns, building large-scale motion time series datasets remains difficult, hindering the development of pre-trained models for human activity analysis. Typically, existing models are trained and tested on the same dataset, leading to poor generalizability across variations in device location, device mounting orientation, and human activity type. In this paper, we introduce UniMTS[1], the first unified pre-training procedure for motion time series that generalizes across diverse device latent factors and activities. Specifically, we employ a contrastive learning framework that aligns motion time series with text descriptions enriched by large language models. This helps the model learn the semantics of time series to generalize across activities. Given the absence of large-scale motion time series data, we derive and synthesize time series from existing motion skeleton data with all-joint coverage. We use spatio-temporal graph networks to capture the relationships across joints for generalization across different device locations. We further design rotation-invariant augmentation to make the model agnostic to changes in device mounting orientations. Our model shows exceptional generalizability across 18 motion time series classification benchmark datasets, outperforming the best baselines by **340%** in the zero-shot setting, **16.3%** in the few-shot setting, and **9.2%** in the full-shot setting.

## 1 Introduction

Recognition of human motion using time series from mobile and wearable devices, such as accelerations and angular velocities, is widely adopted as key context information for various applications from health condition monitoring [4], sports activity analysis [1] to user habit studies [50]. Compared with vision-based approaches, methods based on motion sensor time series offer more energy-efficient and cost-effective solutions with enhanced privacy protection [54], making them preferable.

---

[*]Work unrelated to Amazon.

[1]Code is available on Github: https://github.com/xiyuanzh/UniMTS. Model is available on Hugging Face: https://huggingface.co/xiyuanz/UniMTS.

38th Conference on Neural Information Processing Systems (NeurIPS 2024).

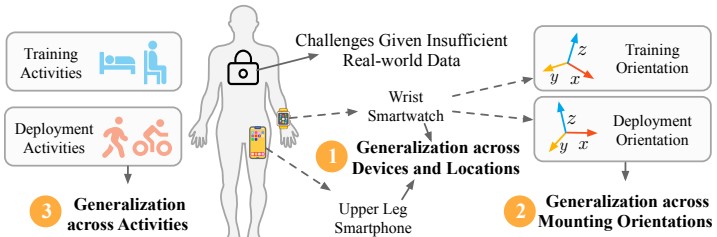

Figure 1: Our framework addresses three key generalization challenges (variation in device location, orientation, and activity) where existing methods fall short.

While valuable, collecting motion time series data at large scale remains challenging due to security or privacy concerns. Labeling motion time series proves even more difficult as such data cannot be easily interpreted by humans for post annotation. This results in data insufficiency that impedes development of supervised learning methods. In other fields such as natural language processing [39, 56] and computer vision [44, 30], pre-trained foundation models have shown remarkable performance in such settings with insufficient data. However, in the motion time series domain, lack of comprehensive datasets and an effective pre-training task makes it difficult to similarly develop pre-trained models that can operate with limited data. Typically, existing models perform training and testing on the same dataset, and struggle to generalize across different datasets given the following three unique challenges within the motion time series problem domain.

We summarize these three unique generalization challenges in Figure 1. First of all, variation in device placement during deployment poses a significant issue; for instance, data from a smartwatch on the wrist vary considerably from data gathered from a smartphone near the upper leg. Therefore, models trained on data from one body location can barely generalize to others during the testing phase. Secondly, devices can experience arbitrary orientations during data collection, making it difficult for models trained on specific device orientations to adapt to new ones during deployment. Thirdly, different motion time series datasets can be focused on different types of human activities. For example, some datasets aim to identify stationary activities such as lying or sitting, while others concentrate on dynamic movements such as walking or cycling. Models trained on specific types of activities typically struggle to generalize to new activities introduced by other datasets.

We introduce UniMTS, the first Unified pre-trained model for Motion Time Series to address all the above three generalization issues, achieving state-of-the-art zero-shot and fine-tuning performance. UniMTS follows a contrastive learning framework that aligns motion time series with LLM-enriched textual descriptions to learn the time series semantics for *activity* generalization. To prepare large-scale motion time series for pre-training, we synthesize these time series based on existing extensive motion skeleton data [19] with comprehensive coverage of different body locations. We model these synthesized time series using graph networks to capture the spatio-temporal relationships across devices for *location* generalization. We further implement rotation-invariant augmentation to ensure the model's robustness to any device *orientation* during testing.

We summarize our primary contributions as follows:

- We introduce the first unified pre-training procedure for motion time series, UniMTS, which successfully generalizes to various device locations, device orientations and activities.
- We design a contrastive learning framework to align motion time series with corresponding semantic meanings for activity generalization. For device location generalization, we propose to synthesize motion time series covering various body locations and model their spatio-temporal correlations using graph convolutional neural networks. We also design rotation-invariant augmentation to make the model agnostic to different device orientations.
- Our pre-trained model demonstrates state-of-the-art performance across 18 real-world motion time series benchmark datasets, notably with performance improvement of **340%** in the zero-shot setting, **16.3%** in the few-shot setting, and **9.2%** in the full-shot setting, compared with the respective best-performing baselines.

## 2 Related Work

**Conventional motion time series classification** approaches train a dedicated classifier for each dataset, and can be categorized into statistical feature extraction methods [15] and deep learning

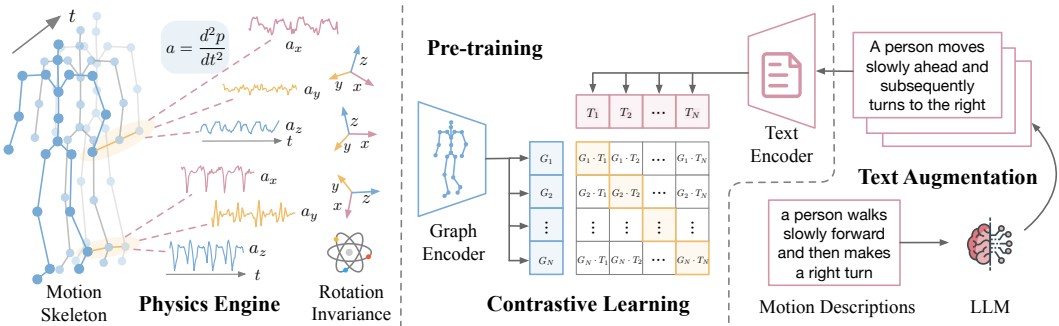

Figure 2: UniMTS pre-training framework: The physics engine computes motion time series for each joint based on motion skeleton data and enhances time series through rotation-invariant augmentation. During pre-training, we adopt contrastive learning to align motion time series encoded by graph convolutional neural networks with corresponding text descriptions augmented by an LLM.

methods, including convolutional neural networks (MA-CNN [45], SenseHAR [23], Rocket [12]), recurrent neural network (DeepConvLSTM [40]), and the attention mechanism based models (AttnSense [36], THAT [29]). Recently, IMUGPT [28, 27] generates motion sequences given activity textual descriptions and trains conventional classification models such as DeepConvLSTM [40]. TimesNet [60], GPT4TS [72] and TEST [52] propose task-general time-series models for multiple tasks including classification. SHARE [71] presents a sequence-to-sequence framework that leverages shared structures of label names. However, these models perform training and testing on the same dataset, and cannot generalize across datasets.

**Self-supervised motion time series representation learning** methods first learn time series representations based on mask reconstruction (TST [68], TARNet [11], LIMU-BERT [62]), contrastive learning (TNC [55], TS-TCC [14], TS2Vec [67], TF-C [70], FOCAL [32], CL-HAR [42], DDLearn [43]) or other self-supervised learning objectives (BioBankSSL [66, 13, 10], Step2Heart [51]). Subsequently, they fine-tune classifier heads for specific downstream tasks. However, the representation learning and fine-tuning phases of these methods generally occur on the same or highly similar datasets, which continues to face challenges in generalization across diverse datasets.

**Pre-trained models for motion time series** are inspired by the recent success of large language or multimodal models. ImageBind [16] and IMU2CLIP [38] leverage recent large vision-language models [44] to learn a joint embedding across multiple modalities including motion time series and text. However, both ImageBind and IMU2CLIP are trained on motion time series collected from head-mounted devices [17], limiting their generalizability across different device locations and orientations. Furthermore, several studies have explored directly applying LLMs for motion time series classification. For example, HARGPT [24] processes raw motion time series through LLMs and incorporates role-play and chain-of-thought strategies for prompting. ContextGPT [3] designs prompt engineering approaches leveraging context information. However, since LLMs are not directly trained on raw motion time series, such methods require extensive context information that is not usually available, and struggle with accurately recognizing complex activities.

**Other related works on motion classification** include multimodal action recognition and domain adaptation methods. Multimodal action recognition such as the Ego4D [17] and Ego-Exo4D [18] benchmarks incorporates video and audio modalities, whereas we focus on a more energy-efficient and challenging scenario of action recognition based purely on motion time series. Domain adaptation methods mostly assume that source and target datasets share the same label names and have the same number of classes, such as cross-user domain adaptation and cross-dataset domain adaptation only for those common classes [22, 35, 21]. We aim for a more generic yet challenging generalization scenario where pre-training and downstream datasets share different label names.

## 3  Method

UniMTS takes a contrastive learning-based approach that aligns paired motion time series with text descriptions to enable activity generalization, as shown in Figure 2. We simulate motion time series from motion skeleton data (Section 3.1) and augment them for orientation generalization (Section 3.2).

We use graph encoder to model the simulated motion time series, capturing correlations among joints to generalize across different device locations (Section 3.3.1). To enhance semantics learning, we use large language models to augment text descriptions (Section 3.3.2).

## 3.1 Physics Engine for Motion Time Series Simulation

Motion skeleton data [19] describe the movements of human skeleton joints over time, containing positions and orientations for each joint. On the other hand, motion time series captured by physical sensors typically measure higher-order data such as accelerations and angular velocities. Consequently, we apply motion equations [65] to synthesize these time series of accelerations and angular velocities from motion skeleton data. More specifically, for each skeleton joint $J_i$, we input both positions $\mathbf{p}_{J_i,\mathcal{G}}$ (mapped from time domain $\mathcal{T}$ to $\mathbb{R}^3$, defined in global frame $\mathcal{G}$), and orientation quaternions $\mathbf{q}_{J_i,\mathcal{GL}}$ (mapped from time domain $\mathcal{T}$ to the Special Orthogonal Group SO(3), defined in Hamilton convention with subscript $\mathcal{GL}$ representing a frame rotation from local frame $\mathcal{L}$ to global frame $\mathcal{G}$). We drop the subscript $\mathcal{G}$ and $\mathcal{GL}$ from here on for simplicity of notation. Based on motion equations [65], we calculate velocities $\mathbf{v}_{J_i}$ and accelerations $\mathbf{a}_{J_i}$ by taking the first and second order derivatives of positions $\mathbf{p}_{J_i}$. These derivatives are then transformed from global frames to local frames using the corresponding orientation sequences $\mathbf{q}_{J_i}$. Similarly, angular velocities $\boldsymbol{\omega}_{J_i}$ are computed by taking the first order derivatives of orientation quaternions $\mathbf{q}_{J_i}$. Mathematically,

$$\mathbf{v}_{J_i}(t) = \mathbf{q}_{J_i}^*(t) \otimes \mathbf{p}'_{J_i}(t) \otimes \mathbf{q}_{J_i}(t), \tag{1}$$

$$\mathbf{a}_{J_i}(t) = \mathbf{q}_{J_i}^*(t) \otimes \mathbf{p}''_{J_i}(t) \otimes \mathbf{q}_{J_i}(t), \tag{2}$$

$$\boldsymbol{\omega}_{J_i}(t) = 2\mathbf{q}_{J_i}^*(t) \otimes \mathbf{q}'_{J_i}(t), \tag{3}$$

where $\otimes$ and $^*$ represent the quaternion multiplication operator and the quaternion conjugate.

Recognizing the inherent presence of noise carried by sensors in practice, the physics engine incorporates Gaussian noise with a zero mean into the simulated data. Representing the above motion time series as $\mathbf{x}_{J_i}(t)$, which can denote either $\mathbf{a}_{J_i}(t)$ (accelerations) or $\boldsymbol{\omega}_{J_i}(t)$ (angular velocities), the noisy time series $\tilde{\mathbf{x}}_{J_i}(t)$ are formulated as

$$\tilde{\mathbf{x}}_{J_i}(t) = \mathbf{x}_{J_i}(t) + \mathbf{n}_{J_i}(t), \mathbf{n}_{J_i}(t) \sim \mathcal{N}(\mathbf{0}, \boldsymbol{\sigma}). \tag{4}$$

## 3.2 Rotation-Invariant Augmentation

A common limitation we have identified from prior studies that leads to their poor generalization is that they fail to consider the impact of latent device orientation factors on the motion time series. For example, end users can potentially wear devices in various orientations, such as with a phone facing towards or against the body in a pocket. Additionally, the software driver API for axis definition can be arbitrarily configured by the developers. For example, the iOS system defines acceleration in an opposite direction compared to the Android system[2]. With the listed risk factors considered, we apply a data augmentation technique to simulate random orientations during pre-training, so that our learned model achieves rotation-invariance during deployment [7, 57, 61]. Specifically, during pre-training, for each iteration we sample a random rotation matrix for each joint $J_i$,

$$\mathbf{R}_{J_i}^\delta \sim \text{Uniform}(\text{SO}(3)), \tag{5}$$

and compute the augmented time series $\hat{\mathbf{x}}_{J_i}^t$ at timesteps $t = 1, 2, \cdots, T$ as

$$\hat{\mathbf{x}}_{J_i}^t = \mathbf{R}_{J_i}^\delta \tilde{\mathbf{x}}_{J_i}^t. \tag{6}$$

During one iteration, the same $\mathbf{R}_{J_i}^\delta$ is consistently applied to $J_i$ for every time series and every timestep $t = 1, 2, \cdots, T$. The rotation-invariant augmentation ensures that the simulated time series are adaptable to any downstream orientation, thereby enhancing the generalization capabilities.

## 3.3 Contrastive Learning

The physics engine generates sufficient motion time series data, which are subsequently encoded by graph networks and aligned with their corresponding text embeddings through contrastive learning.

---

[2] https://github.com/tszheichoi/awesome-sensor-logger/blob/main/CROSSPLATFORM.md

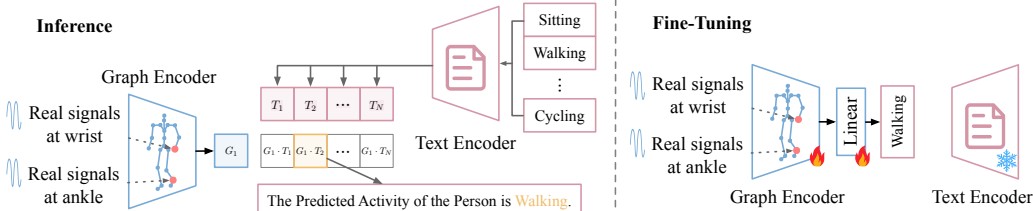

Figure 3: Inference (left) and fine-tuning (right) phases of UniMTS. We assign real signals to the nearest location in the skeleton graph. During inference, we compute the similarity score between the graph embedding and each label candidate, and predict the one with the highest score. During fine-tuning, we freeze the text encoder and update weights of the graph encoder and linear layer.

### 3.3.1 Graph Encoder

To capture the spatio-temporal correlations among different joints over time, we adopt spatio-temporal graph convolutional network [63] as our motion time series encoder. We denote the initial input graph representation as follows,

$$\mathcal{G} = (\mathcal{V} = \{\hat{\mathbf{x}}_{J_i}\}_{i=1}^V, \mathcal{E}_s = \{(\hat{\mathbf{x}}_{J_i}, \hat{\mathbf{x}}_{J_l}) | (J_i, J_l) \in \mathcal{H}\}, \mathcal{E}_t = \{(\hat{\mathbf{x}}_{J_i}^{t-1}, \hat{\mathbf{x}}_{J_i}^t)\}_{i=1,t=2}^{V,T}). \tag{7}$$

Nodes $\mathcal{V}$ contain skeleton joints with features $\mathbf{X} \in \mathbb{R}^{C \times T \times V}$, where $C, T, V$ represent the number of signal channels, temporal steps and joint nodes. Spatial edges $\mathcal{E}_s$ connect adjacent nodes defined by the skeleton structure $\mathcal{H}$ and temporal edges $\mathcal{E}_t$ connect temporally adjacent frames.

In practice, devices may not cover the complete joints but are rather positioned at arbitrary subsets of the complete joints. To simulate this, during each pre-training iteration, we randomly select a subset of joints and mask data from the remaining joints with zeros. We denote the mask at one iteration as $\mathbf{M} \in \mathbb{R}^{C \times T \times V}$, where $\mathbf{M}_i \in \mathbb{R}^{C \times T}$ is $\mathbf{1}$ if joint $J_i$ is selected, and $\mathbf{M}_i = \mathbf{0}$ if joint $J_i$ is masked:

$$\tilde{\mathbf{X}} = \mathbf{X} \odot \mathbf{M}, \tag{8}$$

The graph convolution network $g_\phi$ first computes the spatial output features as

$$\mathbf{X}_{\text{out}} = \Sigma_k^{K_s} \mathbf{\Phi}_k(\tilde{\mathbf{X}}(\mathbf{\Lambda}_k^{-\frac{1}{2}} \mathbf{A}_k \mathbf{\Lambda}_k^{-\frac{1}{2}})), \tag{9}$$

where $K_s$ denotes the spatial kernel size, $\mathbf{A}_k^{il}$ represents whether node $\mathbf{x}_{J_l}$ belongs to the spatial convolution sampling subset $\mathcal{S}_{J_i}^k$ of node $\mathbf{x}_{J_i}$, and $\mathbf{\Lambda}_k^{ii} = \Sigma_l(\mathbf{A}_k^{il}) + \alpha$ represents the normalized diagonal matrix, with $\alpha$ set to 0.001 to prevent empty rows [63, 48]. $\mathbf{\Phi}_k \in \mathbb{R}^{C' \times C \times 1 \times 1}$ represents weights of the $1 \times 1$ convolution operation with $C'$ denoting output channel dimension. Following spatial convolution, we further perform $K_t \times 1$ temporal convolution on the spatial output features $\mathbf{X}_{\text{out}}$, similar to classical convolution operations, where $K_t$ represents the temporal kernel size. The final graph representation $g_\phi(\mathbf{X})$ is derived by averaging features across both spatial and temporal dimensions with a graph average pooling layer at the end.

### 3.3.2 Text Encoder

To increase the diversity of paired text descriptions in the pre-training motion corpus [19], we apply large language models (GPT-3.5) to augment original motion text descriptions with the following prompt template: *The following one or multiple descriptions are describing the same human activities: <motion descriptions>. Generate k paraphrases to describe the same activities.*

We denote the original text descriptions combined with the LLM-augmented ones as $\mathbf{Y}$. We encode them using the same text encoder $f_\theta$ as CLIP [44], utilizing its pre-trained weights for initialization.

### 3.3.3 Training and Inference

During pre-training, we maximize the similarities of paired simulated motion time series and text descriptions through contrastive learning:

$$\mathcal{L}_{ctr} = -\frac{1}{B} \sum_{i=1}^B \log \frac{\exp(\text{sim}(g_\phi(\mathbf{X}_i), f_\theta(\mathbf{Y}_i)))^{\frac{1}{\gamma}}}{\sum_{k=1}^B \exp(\text{sim}(g_\phi(\mathbf{X}_i), f_\theta(\mathbf{Y}_k)))^{\frac{1}{\gamma}}}, \tag{10}$$

where $B, \gamma$ represent batch size and temperature parameter that controls distribution concentrations, and sim represents similarity score computed as inner product:

$$\text{sim}(g_\phi(\mathbf{X}_i), f_\theta(\mathbf{Y}_i)) = \langle g_\phi(\mathbf{X}_i), f_\theta(\mathbf{Y}_i)\rangle. \tag{11}$$

We pre-train the graph and text encoders using simulated motion time series and augmented text descriptions. During inference, we evaluate the model on real-world motion time series, as illustrated in the left part of Figure 3. For the text encoder, we input all label candidates. For the graph encoder, we assign real motion time series to the nearest joint in the skeleton graph and assign zeros to the remaining joints. The random mask $\mathbf{M}$ during pre-training emulates the zero-masking process. We compute the similarity score between the graph embedding with text embedding from each label candidate, and choose the label with the highest similarity score as the predicted activity.

We can further fine-tune the pre-trained model on downstream real-world data, as depicted in the right part of Figure 3. Specifically, we freeze the text encoder $f_\theta$ and update weights of the graph encoder $g_\phi$ followed by a linear classifier $h_\psi$. Following the same process as inference, we assign the real motion time series to the nearest joint in the skeleton graph and assign zeros to the remaining joints to construct the graph input representation $\mathbf{X}$. We fine-tune the model using $\mathbf{X}$ and one-hot encoded labels $\mathbf{z}$ with $D$ classes based on cross-entropy loss, where $\sigma(\cdot)$ represents the softmax operation:

$$\mathcal{L}_{ce} = -\frac{1}{B}\sum_{i=1}^{B}\sum_{j=1}^{D} \mathbf{z}_{ij}\log(\sigma(h_\psi(g_\phi(\mathbf{X}_i)))_j). \tag{12}$$

We report both zero-shot and fine-tuning performance in the subsequent experiment section.

## 4 Experiments

### 4.1 Datasets and Experimental Setting

We simulate motion time series from existing motion skeleton dataset HumanML3D [19], which contain both motion skeleton data and corresponding text descriptions as detailed in Section A.1 in Appendix. We further augment the text descriptions as described in Section 3.3.2.

We evaluate on the most extensive motion time series classification benchmark to date, comprising 18 real-world datasets that cover diverse activities. These datasets are collected from various body locations such as head, chest, back, arm, wrist, waist, hip, leg, knee and ankle. We categorize these datasets into three difficulty levels: (1) easy level (with fewer than 10 activities): Opportunity [47], UCI-HAR [2], MotionSense [37], w-HAR [5], Shoaib [49], HAR70+ [58], RealWorld [53], TNDA-HAR [64]; (2) medium level (with 10 to 20 activities): PAMAP2 [46], USC-HAD [69], Mhealth [4], Harth [33], UT-Complex [50], Wharf [6], WISDM [59], DSADS [1]; (3) hard level (with more than 20 activities): UTD-MHAD [8], MMAct [26]. We provide the specific number of activities for each dataset in Table 1 and Table 2, and detail their collection settings in Section A.2 in Appendix.

We re-sample the real-world test data to the same sampling frequency as the simulation data (20 Hz), and apply normalization to ensure consistency in unit measurements, e.g., standardizing accelerations to $m/s^2$. We pre-train UniMTS using Adam optimizer [25] with a learning rate of $0.0001$ on a single NVIDIA A100 GPU. The pre-training process consumes approximately 13 GB of memory given a batch size of 64. For text augmentation, we prompt GPT-3.5 ("gpt-3.5-turbo") to generate $k = 3$ paraphrases. During each iteration, we randomly generate the mask $\mathbf{M}$ by selecting 1 to 5 joints and mask the remaining joints as zeros. We adopt learnable temperature parameter $\gamma$ initialized from CLIP. We evaluate the models using accuracy, macro-F1 and the top-2 retrieval performance R@2.

### 4.2 Zero-Shot Results

We pre-train UniMTS exclusively on simulated data and evaluate on 18 real-world motion time series classification benchmark datasets. We compare UniMTS against classification models with zero-shot capabilities: ImageBind [16], IMU2CLIP [38], IMUGPT [28] and HARGPT [24]. We also input the 2D visualizations of motion time series to pre-trained vision-language model LLaVA [30] for comparison. We detail the configurations of baselines in Section A.3 in Appendix. As shown in Table 1, UniMTS significantly outperforms all baselines in the zero-shot setting. We also apply the Wilcoxon-signed rank test with Holm's $\alpha$ (5%) following previous works [20, 71]. The Wilcoxon-signed rank test indicates that the improvement of UniMTS compared with all the baselines is

Table 1: Zero-Shot performance. We bold the **best** and underline the second best. UniMTS performs the best compared with both baselines and our model ablations. The last column shows the average performance across 18 datasets with standard deviation.

| Dataset | Metrics | Opportunity | UCI-HAR | MotionSense | w-HAR | Shoaib | HAR70+ | RealWorld | TNDA-HAR | PAMAP2 | USC-HAD | Mhealth | Harth | UT-Complex | Wharf | WISDM | DSADS | UTD-MHAD | MMAct | Average |
|---|---|---|---|---|---|---|---|---|---|---|---|---|---|---|---|---|---|---|---|---|
| Number of Classes | | 4 | 6 | 6 | 7 | 7 | 7 | 8 | 8 | 12 | 12 | 12 | 12 | 13 | 14 | 18 | 19 | 27 | 35 | |
| Level | | Easy | | | | | | | | Medium | | | | | | | | Hard | | Avg |
| ImageBind [16] | Acc | 50.2 | 14.1 | 18.1 | 13.1 | 19.4 | 0.0 | 18.9 | 19.3 | 14.6 | 11.8 | 7.9 | 9.8 | 9.7 | 3.2 | 7.1 | 2.1 | 3.3 | 2.9 | 12.5$_{(11.0)}$ |
| | F1 | 30.0 | 5.0 | 16.4 | 8.6 | 15.5 | 0.0 | 10.1 | 13.7 | 8.1 | 7.3 | 1.7 | 6.1 | 6.3 | 1.9 | 4.5 | 1.2 | 1.6 | 1.5 | 7.8$_{(7.2)}$ |
| | R@2 | **83.6** | 21.4 | 42.8 | 54.1 | 35.5 | 0.2 | 23.9 | 32.8 | 15.6 | 25.0 | 21.3 | 22.5 | 17.8 | 6.8 | 13.7 | 6.9 | 5.1 | 3.7 | 24.0$_{(20.0)}$ |
| IMU2CLIP [38] | Acc | 25.9 | 17.8 | 15.5 | 6.6 | 16.7 | 5.6 | 6.1 | 8.5 | 1.9 | 12.8 | 7.9 | 2.5 | 7.3 | 1.4 | 4.3 | 4.6 | 3.7 | 6.4 | 8.6$_{(6.4)}$ |
| | F1 | 10.3 | 11.4 | 8.6 | 3.3 | 9.2 | 1.8 | 4.5 | 4.2 | 1.1 | 9.3 | 1.3 | 1.1 | 3.4 | 0.7 | 2.7 | 1.1 | 2.1 | 1.6 | 4.3$_{(3.6)}$ |
| | R@2 | 65.5 | 35.1 | 39.1 | 19.7 | 34.6 | 22.2 | 19.7 | 22.4 | 9.8 | 19.8 | 13.4 | 4.8 | 16.3 | 5.5 | 9.6 | 8.3 | 6.5 | 10.1 | 20.1$_{(14.9)}$ |
| IMUGPT [28] | Acc | 10.1 | 1.1 | 11.8 | **67.2** | 12.4 | 0.0 | 16.9 | 14.3 | 8.9 | 6.0 | 9.8 | 4.8 | 11.6 | 2.7 | 8.3 | 7.5 | 3.7 | 2.0 | 11.1$_{(14.4)}$ |
| | F1 | 10.4 | 0.3 | 3.5 | 38.8 | 6.2 | 0.0 | 4.0 | 6.1 | 1.5 | 6.9 | 2.5 | 1.9 | 8.3 | 1.8 | 6.6 | 2.0 | 0.3 | 0.7 | 5.7$_{(8.6)}$ |
| | R@2 | 33.7 | 18.2 | 40.4 | 67.2 | 32.1 | 0.0 | 33.7 | 28.5 | 19.3 | 31.8 | 19.5 | 27.8 | 17.4 | 17.7 | 13.9 | 14.6 | 8.8 | 5.1 | 23.9$_{(14.9)}$ |
| HARGPT [24] | Acc | 28.8 | 15.0 | 11.6 | 4.9 | 21.0 | 34.3 | 12.7 | 13.7 | 11.1 | 9.5 | 10.4 | 28.8 | 7.5 | 5.9 | 5.5 | 5.8 | 3.3 | 2.3 | 12.9$_{(9.2)}$ |
| | F1 | 17.3 | 12.7 | 5.6 | 3.1 | 12.4 | 10.6 | 5.3 | 5.4 | 2.1 | 3.6 | 7.4 | 7.5 | 4.4 | 1.4 | 3.5 | 3.4 | 1.5 | 1.1 | 6.0$_{(4.4)}$ |
| | R@2 | 47.0 | 31.4 | 35.9 | 11.5 | 38.6 | 51.9 | 31.7 | 25.2 | 23.0 | 17.6 | 27.4 | 48.6 | 16.5 | 11.4 | 11.8 | 12.1 | 9.3 | 5.0 | 25.3$_{(14.2)}$ |
| LLaVA [30] | Acc | 40.1 | 16.3 | 22.8 | 0.0 | 16.7 | 10.3 | 16.8 | 12.9 | 10.3 | 11.1 | 18.9 | 16.3 | 2.1 | 3.6 | 5.6 | 5.3 | 3.7 | 4.0 | 12.0$_{(9.4)}$ |
| | F1 | 14.3 | 6.5 | 6.2 | 0.0 | 4.8 | 3.7 | 3.7 | 2.9 | 1.6 | 2.6 | 7.4 | 5.0 | 0.6 | 0.7 | 0.6 | 0.5 | 0.3 | 0.2 | 3.4$_{(3.5)}$ |
| | R@2 | 67.6 | 34.6 | 34.4 | 0.0 | 33.3 | 43.4 | 28.4 | 25.2 | 18.7 | 19.6 | 33.5 | 16.4 | 9.0 | 3.6 | 10.6 | 10.5 | 7.4 | 7.3 | 22.4$_{(16.5)}$ |
| **UniMTS** | Acc | 45.9 | 35.2 | **45.2** | 59.0 | **63.6** | **68.2** | 43.6 | **59.1** | **47.2** | **30.5** | **70.7** | **68.9** | **34.8** | 18.2 | 27.8 | **31.5** | **22.8** | 10.2 | **43.5**$_{(17.9)}$ |
| | F1 | **42.2** | 22.0 | 33.7 | **42.9** | **57.2** | **34.8** | 36.7 | **53.7** | **43.6** | **27.8** | **61.8** | **41.1** | **29.2** | 13.7 | 25.5 | **23.7** | **18.5** | 10.0 | **34.3**$_{(14.2)}$ |
| | R@2 | 80.0 | 53.1 | 57.2 | 60.7 | **82.1** | **86.6** | 64.0 | **77.5** | **63.2** | 45.4 | **78.7** | **85.0** | **44.2** | 38.6 | **47.1** | **46.0** | **32.6** | **18.7** | **58.9**$_{(19.3)}$ |
| w/o rot aug | Acc | 37.7 | 18.6 | 25.1 | 36.1 | 19.4 | 53.4 | **55.5** | 35.6 | 32.5 | 20.7 | 27.4 | 59.4 | 9.9 | 3.6 | 13.5 | 21.5 | 13.5 | 4.3 | 27.1$_{(16.3)}$ |
| | F1 | 30.4 | 8.2 | 11.2 | 26.4 | 13.5 | 27.7 | **41.1** | 29.6 | 28.3 | 10.5 | 24.0 | 20.5 | 4.9 | 4.8 | 10.6 | 13.4 | 7.2 | 4.7 | 17.6$_{(10.7)}$ |
| | R@2 | 74.3 | 40.1 | **64.3** | 52.5 | 40.1 | 76.5 | **76.3** | 54.4 | 48.8 | 29.7 | 49.4 | 61.2 | 28.5 | 6.8 | 23.3 | 34.0 | 32.1 | 7.6 | 44.4$_{(20.8)}$ |
| w/o text aug | Acc | **52.7** | 36.3 | 43.4 | 57.4 | 55.6 | 61.0 | 40.7 | 40.9 | 38.4 | 29.6 | 59.2 | 62.8 | 30.3 | **28.2** | **31.3** | 29.3 | 6.5 | **12.6** | 39.8$_{(15.8)}$ |
| | F1 | 39.4 | 21.3 | **34.7** | 41.4 | 49.5 | 32.8 | 29.7 | 32.7 | 33.9 | 21.6 | 49.0 | 26.7 | 21.2 | **19.6** | **27.2** | 22.2 | 4.7 | **10.3** | 28.8$_{(11.6)}$ |
| | R@2 | 70.9 | 39.6 | 63.7 | 57.4 | 78.7 | 77.4 | 60.4 | 63.5 | 48.8 | **49.1** | 63.4 | 77.9 | 41.4 | **43.2** | 45.1 | 41.7 | 10.7 | 23.2 | 53.1$_{(18.1)}$ |
| w/o graph | Acc | 41.4 | **37.6** | 18.9 | 23.0 | 50.9 | 18.8 | 42.7 | 33.8 | 30.0 | 22.4 | 28.7 | 34.8 | 23.4 | 0.5 | 12.4 | 19.8 | 1.9 | 11.2 | 25.1$_{(13.4)}$ |
| | F1 | 20.5 | **28.5** | 21.6 | 17.9 | 38.6 | 9.1 | 23.7 | 28.9 | 23.6 | 23.0 | 19.3 | 15.3 | 11.5 | 1.1 | 8.5 | 16.0 | 2.7 | 7.7 | 17.6$_{(9.5)}$ |
| | R@2 | 63.7 | **66.7** | 54.2 | 34.4 | 66.1 | 35.3 | 51.1 | 62.1 | 41.8 | 41.7 | 43.9 | 51.6 | 35.6 | 8.2 | 23.7 | 29.1 | 4.7 | 18.0 | 40.7$_{(18.4)}$ |

statistically significant, with p-values significantly lower than 0.05 (e.g., p-value = $8 \times 10^{-6}$ for ImageBind, which has the highest F1 score among the baselines).

Compared with UniMTS, ImageBind and IMU2CLIP are trained on data from single location (head-mounted devices), limiting their generalization to data collected from other locations. IMUGPT struggles to generalize across datasets featuring different activities and requires individual training for each downstream dataset. Both HARGPT and LLaVA focus on simple and easily distinguishable activities as these language or vision models are not originally trained on motion time series, and they also require careful prompt designs. Another limitation for all the above models is that they do not generalize across device orientations. In contrast, UniMTS shows remarkable generalizability to various downstream device locations, orientations and activities, achieving state-of-the-art performance. We also compare with a few ablations of UniMTS as illustrated in the Ablation Study section.

### 4.3 Few-Shot Fine-tuning Results

Apart from the zero-shot setting, we provide a few real samples for each activity and fine-tune UniMTS and the baselines. More specifically, we provide 1, 2, 3, 5, 10 samples for each activity and compare UniMTS against ImageBind [16], IMU2CLIP [38], IMUGPT [28], GPT4TS [72], BioBankSSL [66] and a randomly initialized model with the same model architecture as UniMTS (referred to as Random). We report both the mean and the standard deviation in Figure 4. UniMTS also demonstrates state-of-the-art performance in the few-shot fine-tuning setting, showing the effectiveness of pre-training. Following the same Wilcoxon-signed rank test as in the zero-shot setting, we observe p-values far below 0.05 (e.g., p-value = $2 \times 10^{-25}$ for the best-performing baseline ImageBind), indicating the statistical significance of our improvement.

### 4.4 Full-Shot Results

We also compare the full-shot performance where UniMTS and the baselines are fine-tuned or trained using all the available training samples of the downstream datasets. We compare UniMTS with pre-trained models (ImageBind [16], IMU2CLIP [38]), self-supervised models (TST [68], TARNet [11],

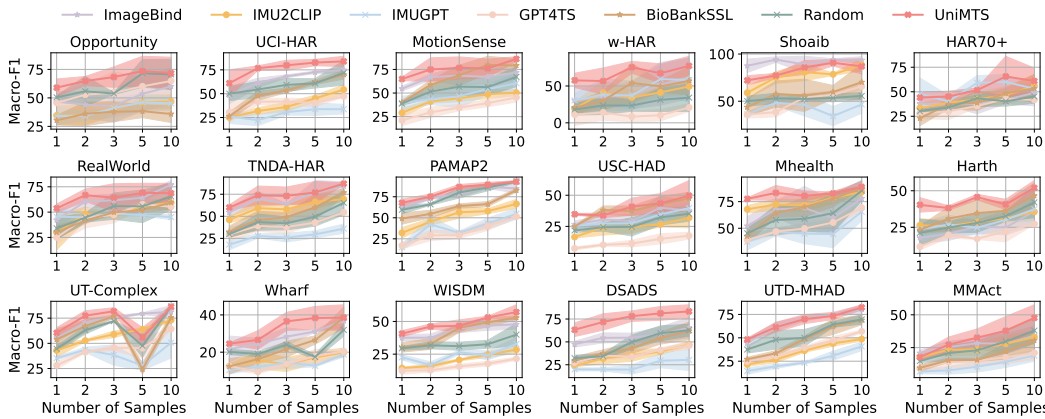

Figure 4: Few-shot fine-tuning results. UniMTS consistently outperforms both baselines and our model ablation. We repeat 3 runs and report both mean and standard deviation.

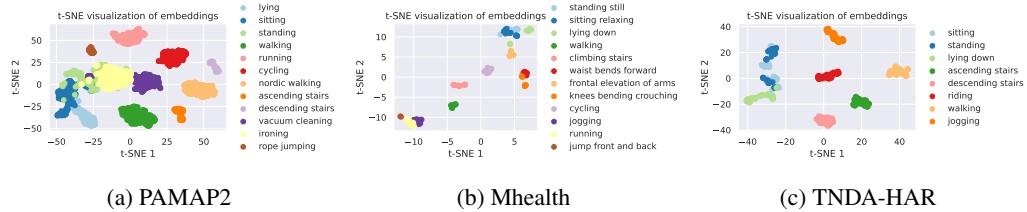

(a) PAMAP2          (b) Mhealth          (c) TNDA-HAR

Figure 5: T-SNE visualizations show that signal clusters align with their semantic meanings.

TS2Vec [67], BioBankSSL [66]), and conventional models (DeepConvLSTM [40], MA-CNN [45], XGBoost [9], THAT [29], IMUGPT [28], TimesNet [60], GPT4TS [72], SHARE [71]). Baselines are detailed in Section A.3 in Appendix. We also compare pre-trained UniMTS with a randomly initialized UniMTS (referred to as Random). As shown in Table 2, UniMTS also demonstrates state-of-the-art performance in the full-shot setting, outperforming pre-trained, self-supervised and conventional models. Due to space limit, we report baselines before 2021 in Table 3 in Appendix. Following the same Wilcoxon-signed rank test, we observe p-values far below 0.05 (e.g., p-value = 0.018 for the best-performing baseline), indicating the statistical significance of our improvement. UniMTS also demonstrates space and time efficiency, as detailed in Section A.5 in Appendix.

## 4.5 Ablation Study

In the zero-shot setting, we compare UniMTS with a few ablations by removing rotation-invariant augmentation (w/o rot aug), removing text augmentation (w/o text aug) and by replacing the graph encoder with a CNN-based encoder that directly concatenates joints without modeling their spatial relationships (w/o graph). We can observe in Table 1 that the performance declines after removing each of the above components, verifying their respective importance in improving generalization across locations (graph encoder), orientations (rotation-invariant augmentation) and activities (text augmentation). We also compare the pre-trained UniMTS with randomly initialized UniMTS in both few-shot and full-shot settings. As shown in Figure 4 and Table 2, pre-trained UniMTS consistently outperforms randomly initialized UniMTS, highlighting the benefits of pre-training.

## 4.6 Case Study

**UniMTS's time series embeddings align with corresponding semantic meanings.** As shown in Figure 5, the t-SNE visualizations of UniMTS's time series embeddings form distinguishable clusters that align with their semantic meanings. Notably, UniMTS is only pre-trained on the simulated data but its embeddings for real-world data closely align with the semantic space, which again demonstrates our model's zero-shot generalization due to contrastive learning. For example, in Figure 5a, stationary activities such as lying and sitting group together; light-movement activities such as standing, ironing, and vacuum cleaning are close to each other; while high-intensity activities such as running and cycling cluster closer in the embedding space.

Table 2: Full-Shot performance. We bold the **best** and underline the second best. UniMTS performs the best compared with both pre-trained, self-supervised and conventional models. The last column shows the average performance across 18 datasets with standard deviation.

| Dataset | Metrics | Opportunity | UCI-HAR | MotionSense | w-HAR | Shoaib | HAR70+ | RealWorld | TNDA-HAR | PAMAP2 | USC-HAD | Mhealth | Harth | UT-Complex | Wharf | WISDM | DSADS | UTD-MHAD | MMAct | Average |
|---|---|---|---|---|---|---|---|---|---|---|---|---|---|---|---|---|---|---|---|---|
| Number of Classes | | 4 | 6 | 6 | 7 | 7 | 7 | 8 | 8 | 12 | 12 | 12 | 12 | 13 | 14 | 18 | 19 | 27 | 35 | |
| Level | | | | | Easy | | | | | | | | Medium | | | | | Hard | | Avg |
| ImageBind [16] | Acc | 82.8 | 90.9 | 94.0 | 95.1 | 98.5 | 91.8 | 83.8 | 95.0 | 94.3 | 52.1 | 82.9 | 93.0 | 97.0 | 71.4 | 77.2 | 90.0 | 74.0 | 65.8 | 85.0$_{(12.2)}$ |
| | F1 | 84.6 | 90.7 | 93.1 | 61.9 | 98.5 | 67.7 | **86.1** | 94.9 | 94.6 | 52.2 | 83.2 | 62.7 | 96.9 | 43.3 | 76.8 | 89.3 | 71.6 | 58.3 | 78.1$_{(16.5)}$ |
| IMU2CLIP [38] | Acc | 71.9 | 87.1 | 79.3 | 88.5 | 98.2 | 87.8 | 79.9 | 97.1 | 90.7 | 55.7 | 94.5 | 93.9 | 90.6 | 52.3 | 62.7 | 88.7 | 62.8 | 66.9 | 80.5$_{(14.3)}$ |
| | F1 | 70.0 | 87.0 | 77.2 | 75.1 | 98.1 | 64.4 | 80.5 | 97.1 | 91.2 | 52.0 | 95.0 | 65.7 | 90.2 | 35.6 | 62.8 | 88.2 | 59.6 | 60.3 | 75.0$_{(17.0)}$ |
| TST [68] | Acc | **89.1** | 91.6 | 91.4 | 75.4 | 73.5 | 95.8 | 68.9 | 94.1 | 88.9 | 66.2 | 86.0 | 98.8 | 92.9 | 59.1 | 70.2 | 80.0 | 61.4 | 54.5 | 79.9$_{(13.6)}$ |
| | F1 | **91.1** | 91.4 | 89.6 | 79.3 | 72.3 | 62.9 | 67.2 | 94.1 | 89.8 | 60.9 | 83.8 | 72.7 | 92.8 | 34.5 | 69.1 | 76.1 | 59.6 | 48.4 | 74.2$_{(16.3)}$ |
| TARNet [11] | Acc | 82.6 | 91.1 | 72.0 | 78.7 | 67.9 | 97.2 | 60.9 | 92.5 | 92.4 | 57.6 | 82.9 | 94.7 | 94.4 | 63.2 | 58.5 | 30.7 | 78.1 | 56.7 | 75.1$_{(17.5)}$ |
| | F1 | 83.4 | 91.2 | 70.0 | 41.0 | 66.1 | 76.9 | 54.4 | 92.4 | 90.6 | 48.7 | 77.5 | 47.8 | 94.4 | 38.1 | 58.5 | 23.0 | 75.5 | 55.9 | 65.9$_{(20.5)}$ |
| TS2Vec [67] | Acc | 80.7 | 92.1 | 94.2 | **100.0** | 87.4 | **98.4** | 74.6 | 95.9 | 91.8 | 56.1 | 93.3 | 98.2 | 98.5 | 73.2 | 69.5 | 84.3 | 80.0 | 53.4 | 84.5$_{(13.9)}$ |
| | F1 | 83.3 | 92.1 | 93.9 | **100.0** | 87.2 | **88.1** | 68.8 | 95.9 | 92.9 | 53.6 | 93.8 | 58.3 | 98.5 | 47.9 | 68.8 | 82.3 | 76.7 | 53.4 | 79.8$_{(16.7)}$ |
| BioBankSSL [66] | Acc | 85.9 | **92.7** | 99.4 | 93.4 | 99.1 | 93.2 | 75.5 | 89.4 | 87.3 | 65.8 | 95.1 | 98.0 | 96.4 | 81.4 | **83.3** | 65.8 | 80.5 | 62.5 | 85.8$_{(11.5)}$ |
| | F1 | 86.1 | **92.9** | 99.2 | 81.3 | 99.1 | 85.9 | 68.0 | 89.6 | 89.3 | 72.6 | 94.0 | 68.1 | 96.4 | 62.7 | 82.9 | 83.1 | 78.4 | 58.4 | 82.7$_{(12.1)}$ |
| THAT [29] | Acc | 83.1 | 86.8 | 87.9 | 77.1 | 92.3 | 95.8 | 67.7 | 97.7 | 96.5 | 53.9 | 89.0 | 97.6 | 86.9 | 60.0 | 60.3 | 82.1 | 71.2 | 55.0 | 80.1$_{(14.7)}$ |
| | F1 | 84.5 | 86.7 | 86.5 | 65.0 | 92.2 | 73.9 | 62.9 | 97.7 | 96.7 | 55.0 | 89.9 | 71.9 | 87.0 | 29.5 | 60.9 | 79.0 | 70.0 | 52.3 | 74.5$_{(17.5)}$ |
| IMUGPT [28] | Acc | 84.8 | 87.0 | 86.2 | 85.3 | 61.7 | 98.3 | 62.2 | 87.7 | 85.6 | 41.1 | 71.3 | 98.3 | 86.5 | 54.6 | 67.8 | 71.9 | 57.7 | 51.9 | 74.4$_{(16.3)}$ |
| | F1 | 84.9 | 87.0 | 85.2 | 48.8 | 61.8 | 78.0 | 56.0 | 87.5 | 83.8 | 42.4 | 63.6 | 65.2 | 85.8 | 24.6 | 67.4 | 71.0 | 53.2 | 49.0 | 66.4$_{(17.7)}$ |
| TimesNet [60] | Acc | 80.0 | 88.5 | 90.5 | 88.5 | 82.4 | 97.5 | 65.4 | 93.2 | 88.0 | 58.3 | 81.7 | 97.7 | 92.7 | 41.4 | 67.4 | 77.3 | 62.3 | 50.9 | 78.0$_{(16.2)}$ |
| | F1 | 82.4 | 88.6 | 88.3 | 79.5 | 82.6 | 77.2 | 62.2 | 93.1 | 88.0 | 48.7 | 79.1 | 61.4 | 92.8 | 30.2 | 67.1 | 78.5 | 61.0 | 45.7 | 72.6$_{(17.3)}$ |
| GPT4TS [72] | Acc | 83.6 | 88.4 | 92.3 | 70.5 | 73.5 | 97.8 | 66.5 | 95.7 | 92.6 | 61.7 | 87.8 | 97.2 | 95.3 | 51.8 | 63.8 | 79.9 | 66.1 | 52.6 | 78.7$_{(15.2)}$ |
| | F1 | 86.1 | 88.5 | 92.1 | 54.8 | 74.1 | 76.6 | 56.6 | 95.7 | 93.4 | 58.9 | 85.2 | 58.2 | 95.2 | 34.4 | 64.5 | 78.0 | 63.7 | 48.8 | 72.5$_{(17.7)}$ |
| SHARE [71] | Acc | 89.0 | 92.2 | 99.6 | 96.7 | 77.5 | 97.7 | 66.0 | 95.9 | 94.1 | **72.9** | **97.0** | **99.1** | 98.7 | 63.2 | 77.0 | 92.0 | 73.0 | 62.1 | 85.8$_{(13.2)}$ |
| | F1 | 90.2 | 92.1 | 99.6 | 85.5 | 78.0 | 77.5 | 57.1 | 95.8 | 94.8 | 66.7 | 97.1 | 74.0 | 98.8 | 40.6 | 76.5 | 91.9 | 69.3 | 56.5 | 80.1$_{(16.5)}$ |
| **UniMTS** | Acc | 88.2 | 92.1 | **99.8** | 98.4 | **99.7** | 96.9 | **84.0** | **99.6** | **98.0** | 58.3 | **97.0** | 98.0 | **99.1** | **82.7** | 81.3 | **94.3** | **85.6** | **77.1** | **90.6**$_{(10.6)}$ |
| | F1 | 89.1 | 92.2 | **99.8** | 98.8 | **99.7** | 87.9 | 81.2 | **99.6** | **98.1** | 63.1 | **97.3** | **86.7** | **99.2** | **51.7** | 80.9 | **94.2** | **83.2** | **71.7** | **87.5**$_{(13.4)}$ |
| Random | Acc | 87.7 | 89.3 | 95.5 | 95.1 | 66.1 | 98.1 | 74.2 | 98.7 | 96.2 | 48.5 | 82.9 | 98.6 | 98.5 | 76.8 | 79.4 | 90.3 | 74.4 | 73.9 | 84.7$_{(13.5)}$ |
| | F1 | 88.9 | 89.4 | 95.5 | 60.1 | 65.0 | 85.9 | 74.0 | 98.7 | 96.8 | 51.9 | 78.2 | 69.1 | 98.5 | 50.8 | 79.3 | 90.1 | 69.7 | 71.6 | 78.5$_{(15.1)}$ |

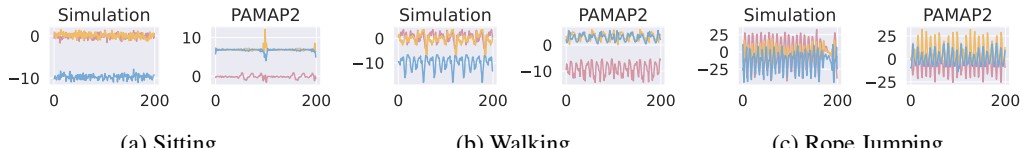

(a) Sitting      (b) Walking      (c) Rope Jumping

Figure 6: Simulated motion time series closely resemble patterns of the real PAMAP2 time series.

**UniMTS generalizes well to new activities unseen in pre-training.** To show UniMTS's capability to generalize to new activities not seen during pre-training, we visualize the *zero-shot* performance for some example new activities in Figure 7. Compared with the best-performing baseline Image-Bind, UniMTS shows significant performance improvement on these previously unseen activities, verifying the effectiveness of semantic generalization via contrastive learning.

**Simulated data from our physics engine closely resemble real signal patterns.** In Figure 6, we compare some example simulated data alongside their real-world counterparts. We show three example activities of different intensity levels (i.e., sitting, walking, rope jumping), where both simulated

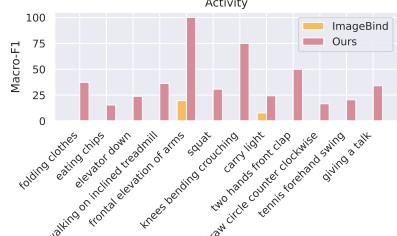

Figure 7: UniMTS shows significant performance improvement compared with the best baseline when evaluated on new activities not seen.

data and real-world data are near the wrist. We can observe that the patterns in simulated data closely resemble those of real data, in terms of both magnitude and frequency. Although the tri-axial distributions might differ, these variations are mostly due to orientation differences and are effectively managed by our rotation-invariant augmentation. We observe similar patterns between simulated data and real data for other device locations, as shown in Section A.4 in Appendix.

# 5 Conclusion and Discussion

**Conclusion**. In this paper, we present the first unified pre-training procedure, UniMTS, for motion time series classification. Our model is pre-trained only on physics-simulated data, and yet demonstrates remarkable generalization across diverse real-world motion time series datasets featuring different device locations, orientations and activities. The simulated data with all-joint coverage are augmented for rotation invariance and modeled by a graph encoder, improving generalization across various device factors. During pre-training, contrastive learning aligns time series with their semantic meanings to improve generalization across activities. Extensive evaluation in zero-shot, few-shot and full-shot settings consistently demonstrates the state-of-the-art performance of UniMTS.

**Limitation and Future Work**. We acknowledge a few limitations which we leave as future work. (1) Simulated motion time series can only be approximations of real signals, which are usually collected near – rather than directly on – the body joints. For example, sensors on smartwatches collect data near the wrist, not on the wrist joint itself. We plan to incorporate random offset vectors to better simulate real-world signal variations near joints. (2) While our framework effectively addresses the classification task, we intend to extend its applicability to other motion time series tasks such as inertial navigation. (3) Our current pre-training utilizes existing motion datasets, and we plan to enrich our pre-training corpus with additional motion data extracted from large-scale video-based pose estimation. (4) We also plan to integrate our model with efficient inference optimization techniques such as quantization, pruning and distillation for deployment on edge devices.

**Broad Impact**. UniMTS is the first pre-trained motion time series classification model that generalizes to diverse downstream datasets, irrespective of device locations, orientations and activities. The primary societal concern centers around privacy as motion time series might reveal personal information, so we ensure strict privacy controls at the earliest stages of model development by pre-training exclusively on synthetic data. With UniMTS's state-of-the-art performance in zero-shot, few-shot and full-shot settings, we believe it would bring broad, positive impact to the community.

# 6 Acknowledgement

Our work is supported in part by ACE, one of the seven centers in JUMP 2.0, a Semiconductor Research Corporation (SRC) program sponsored by DARPA.

Our work is also sponsored in part by NSF CAREER Award 2239440, NSF Proto-OKN Award 2333790, as well as generous gifts from Google, Adobe, and Teradata. Any opinions, findings, and conclusions or recommendations expressed herein are those of the authors and should not be interpreted as necessarily representing the views, either expressed or implied, of the U.S. Government. The U.S. Government is authorized to reproduce and distribute reprints for government purposes not withstanding any copyright annotation hereon.

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

# A   Appendix / supplemental material

## A.1   Pre-training Datasets

HumanML3D [19] is a large-scale motion skeleton data consisting of 14,616 3D human motion skeletons spanning 28.59 hours. The average motion skeleton sequence length is 7.1 seconds. Paired with each motion skeleton sequence there is an average of 3 textual descriptions, resulting in a total of 44,970 textual descriptions with a vocabulary size of 5,371. The average and median lengths of these descriptions are 12 and 10 words. We further augment the textual descriptions using large language models as described in Section 3.3.2. All motion skeletons follow the skeleton structure of SMPL [34] with 22 joint nodes. Figure 8 provides an example skeleton of "a person waves his hands".

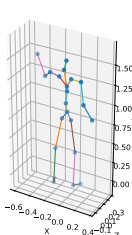

Figure 8: Skeleton of "waving hands".

We also tried to incorporate additional motion skeleton datasets into pre-training, such as KIT-ML [41] and NTU RGB+D 120 [31]. However, these data are relatively less diverse in terms of both motion skeletons and textual descriptions. We did not observe performance improvement from adding them, and therefore use HumanML3D as our primary pre-training corpus.

## A.2   Downstream Evaluation Datasets

We detail the information for each downstream real-world evaluation dataset as follows.

**Opportunity** [47] contains data collected from back, upper arms and lower arms, and features multiple sets of activities. We aim to predict the modes of locomotion such as standing and walking.

**UCI-HAR** [2] collects motion data from a smartphone located on the subject's waist. The subject performs daily activities such as walking upstairs and walking downstairs.

**MotionSense** [37] collects data from a smartphone in the participant's front pocket, featuring daily activities such as sitting and jogging.

**w-HAR** [5] contains motion time series data collected from the ankle. It captures daily physical activities such as jumping and lying down.

**Shoaib** [49] contains daily activities such as biking. Each participant is equipped with five smartphones on five positions: right jean's and left jean's pockets, belt, right upper arm and right wrist.

**HAR70+** [58] tracks activities such as shuffling for older adult subjects. The motion time series are collected from the right front thigh and the lower back.

**RealWorld** [53] records daily activities such as climbing stairs from multiple body positions including chest, forearm, head, shin, thigh, upper arm, and waist.

**TNDA-HAR** [64] collects static as well as periodic daily activities such as cycling, from devices located at multiple body positions such as wrist, ankle and back.

**PAMAP2** [46] monitors physical activities such as ironing, vacuum cleaning and rope jumping using devices located on the wrist, chest and ankle.

**USC-HAD** [69] records daily activities such as sleeping and taking the elevator with devices attached to the subject's front right hip.

**Mhealth** [4] comprises body motion for common activities such as waist bending forward, frontal elevation of arms and knees bending. Devices are placed on the user's chest, right wrist and left ankle.

**Harth** [33] records data in a free-living setting with devices located at the right thigh and lower back.

**UT-Complex** [50] contains different smartphone sensor data such as typing, drinking coffee and giving a talk, with devices positioned at wrist and pocket positions.

**Wharf** [6] records activities from wrist-worn devices, such as combing hair and getting up bed.

**WISDM** [59] collects diverse daily activities such as brushing teeth, eating soup, playing balls, and folding clothes, using data from the smartphone in the pocket and smartwatch on hand.

Table 3: Full-Shot performance on additional baselines before 2021. We bold the **best** results. The last column shows the average performance across 18 datasets with standard deviation.

| Dataset | Metrics | Opportunity | UCI-HAR | MotionSense | w-HAR | Shoaib | HAR70+ | RealWorld | TNDA-HAR | PAMAP2 | USC-HAD | Mhealth | Harth | UT-Complex | Wharf | WISDM | DSADS | UTD-MHAD | MMAct | Average |
|---|---|---|---|---|---|---|---|---|---|---|---|---|---|---|---|---|---|---|---|---|
| Number of Classes | | 4 | 6 | 6 | 7 | 7 | 7 | 8 | 8 | 12 | 12 | 12 | 12 | 13 | 14 | 18 | 19 | 27 | 35 | |
| Level | | Easy | | | | | | | | Medium | | | | | | | | Hard | | Avg |
| DeepCL [40] | Acc | 84.4 | 86.9 | 86.2 | 93.4 | 62.4 | **97.7** | 69.9 | 89.6 | 82.7 | 45.2 | 73.8 | 95.5 | 88.0 | 46.8 | 69.0 | 69.6 | 54.9 | 50.2 | 74.8(16.7) |
| | F1 | 86.4 | 87.0 | 84.5 | 67.1 | 63.4 | **92.2** | 60.3 | 89.3 | 77.4 | 46.4 | 75.8 | 56.7 | 88.2 | 25.9 | 69.0 | 66.1 | 53.0 | 46.9 | 68.6(17.8) |
| MA-CNN [45] | Acc | 82.9 | 89.8 | 92.0 | 67.2 | 93.2 | 86.1 | 73.9 | 96.0 | 94.7 | 37.3 | 78.7 | 97.9 | 93.8 | 20.9 | 51.5 | 84.5 | 48.8 | 28.9 | 73.2(24.2) |
| | F1 | 84.8 | 89.5 | 91.7 | 51.7 | 93.1 | 59.8 | 71.2 | 96.0 | 95.2 | 35.3 | 70.7 | 53.2 | 94.0 | 17.2 | 51.8 | 83.9 | 48.5 | 18.4 | 67.0(25.5) |
| XGBoost [9] | Acc | 83.1 | 90.2 | 92.9 | 68.9 | 94.8 | **97.7** | 78.2 | 93.2 | 93.9 | 47.8 | 79.9 | 97.2 | 96.6 | 52.3 | 66.6 | 80.5 | 61.9 | 53.0 | 79.4(16.5) |
| | F1 | 85.1 | 90.1 | 91.6 | 56.1 | 94.7 | 77.5 | 77.6 | 93.2 | 93.9 | 47.7 | 74.1 | 64.4 | 96.7 | 30.2 | 66.0 | 79.4 | 60.3 | 51.4 | 73.9(18.6) |
| **UniMTS** | Acc | **88.2** | **92.1** | **99.8** | **98.4** | **99.7** | 96.9 | **84.0** | **99.6** | **98.0** | **58.3** | **97.0** | **98.0** | **99.1** | **82.7** | **81.3** | **94.3** | **85.6** | **77.1** | **90.6**(10.6) |
| | F1 | **89.1** | **92.2** | **99.8** | **98.8** | **99.7** | 87.9 | **81.2** | **99.6** | **98.1** | **63.1** | **97.3** | **86.7** | **99.2** | **51.7** | **80.9** | **94.2** | **83.2** | **71.7** | **87.5**(13.4) |

**DSADS** [1] comprises daily and sports activities such as exercising and rowing. Multiple devices are positioned at the torso, right arm, left arm, right leg, and left leg.

**UTD-MHAD** [8] contains diverse activities such as swiping arms, hand clapping, throwing, arm crossing, drawing and squatting. The devices are worn on the subject's right wrist or the right thigh depending on whether the action is mostly an arm or a leg type of action.

**MMAct** [26] presents a large-scale activity dataset covering a wide range of daily life activities such as carrying, talking on phone and falling. Devices recording motion time series include a smartwatch as well as a smartphone inside the pocket of the subject's pants.

## A.3 Baselines

We detail the baseline settings as follows.

**ImageBind** [16]: We employ the pre-trained weights from "imagebind_huge" for zero-shot evaluation. During fine-tuning, we add a linear layer to map ImageBind embeddings to the number of activity classes. We fine-tune both ImageBind and the linear layer during fine-tuning, which performs better than simply tuning the linear layer.

**IMU2CLIP** [38]: The pre-trained weights of IMUCLIP are not released. Therefore, we first follow their pre-training implementation[3] to pre-train on Ego4D datasets [17]. During fine-tuning, we add a linear layer after IMU2CLIP embeddings and fine-tune both IMU2CLIP and the linear layer.

**IMUGPT** [28]: We choose DeepConvLSTM as the backbone model, which shows the best performance as reported in their original paper. We remove the supervised distribution calibration phase, which relies on labeled downstream data and conflicts with the zero-shot setting objectives.

**HARGPT** [24]: The method directly prompts large language models to classify motion time series. We down-sample motion time series to 10 Hz as used in their paper and follow their prompt template.

**LLaVA** [30]: We visualize motion time series as 2D plots and use these visualizations as input for the pre-trained model of "llava-v1.5-7b".

**TST** [68]: This is a Transformer-based representation learning framework with several downstream tasks including multivariate time-series classification. We follow the framework to first pre-train the Transformer model in an unsupervised fashion and then fine-tune the pre-trained model on the downstream classification task.

**TARNet** [11]: The model proposes task-aware representation learning that reconstructs important timestamps guided by self-attention score distribution from end-task training. We jointly train the reconstruction task and the classification task.

---

[3]https://github.com/facebookresearch/imu2clip

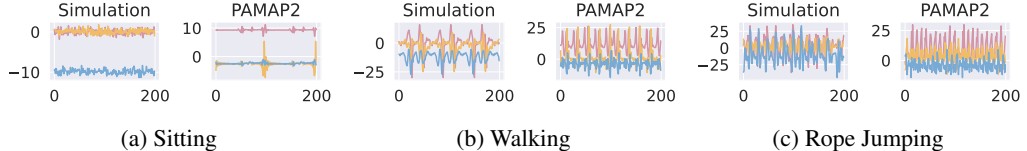

(a) Sitting             (b) Walking             (c) Rope Jumping

Figure 9: Simulated motion time series show similar patterns as real PAMAP2 time series.

**TS2Vec** [67]: The method performs contrastive learning to learn contextual representations of time series. We follow their implementation to first apply contrastive learning and then train a linear regression model for each dataset.

**BioBankSSL** [66]: The paper proposes a pre-trained model for human activity recognition using a large-scale UK Biobank wrist accelerometer dataset with multi-task self-supervised learning.

**DeepConvLSTM** [40] (shown as "DeepCL" in Table 3 due to space limit): The model applies convolutional layers to automatically learn feature representations and further applies LSTM to capture the temporal dependencies between their activations.

**MA-CNN** [45]: The model first extracts preliminary features for each motion time series modality through its own dedicated convolutional layers, then the extracted intra-modality features are combined through fully-connected layers for motion time series classification.

**XGBoost** [9]: This is a scalable end-to-end machine learning system for tree boosting, which has been widely recognized in machine learning and time series analysis.

**THAT** [29]: The model proposes a two-stream convolution augmented human activity transformer which captures both time-over-channel and channel-over-time features in a two-stream structure.

**TimesNet** [60]: This is a task-general backbone for time series analysis including classification by modeling the multi-periodicity and extracting temporal variations.

**GPT4TS** [72]: This is also a task-general framework that includes time series classification. The model is based on a frozen pre-trained language model, so we also adopt it as a few-shot fine-tuning baseline. However, the method is not suitable for zero-shot evaluation, as it requires training a separate classifier head for each downstream dataset and does not generalize across activities.

**SHARE** [71]: This is a sequence-to-sequence model that contains an encoder to extract motion time series features, as well as a decoder to generate label name sequences to capture label semantics.

## A.4 Simulated Data

In addition to the simulated data for wrist as shown in Figure 6, we present more examples for other device locations such as ankle in Figure 9. We observe consistently similar patterns between simulated motion time series and real PAMAP2 data across activities of various intensity levels, ranging from sitting to walking and rope jumping.

## A.5 Efficiency Analysis

For *space complexity*, the graph encoder of UniMTS contains only 4.94M parameters, which is significantly smaller compared with the 18.69M used in the IMU encoder of the best existing baseline ImageBind. For *time complexity*, fine-tuning of UniMTS is also efficient. On one example dataset of UCI-HAR, full-shot fine-tuning of UniMTS takes approximately 1.3 minutes to converge while it takes approximately 9.8 minutes for ImageBind to converge. Moreover, we have run a power estimate assuming 0.1Hz cadence (i.e., 10-second window size), and it takes approximately 22.64 mW to run the whole graph model on an eNPU (embedded Neural Processing Unit), which is much smaller than ImageBind IMU encoder's power consumption of approximately 702 mW. Therefore, UniMTS is efficient for real-world applications and suitable to be deployed on edge devices.

