# OpenReview forum: "UniMTS: Unified Pre-training for Motion Time Series"
_NeurIPS.cc/2024/Conference — NeurIPS 2024 poster_

### Official Review · Reviewer_s9ZD · 2024-07-03

**Soundness:** 3
**Presentation:** 3
**Contribution:** 3
**Rating:** 6
**Confidence:** 4

**Summary:**

Overall, this is a paper about human activity recognition from motion time series data. Authors design a unified framework for data generation, pre-training, and evaluation. The data used for pre-training is generated by motion skeleton data. LLM is also involved in text augmentation. Experimental results on 18 datasets demonstrate the superior performance of this paper.

**Strengths:**

[1] This paper is well-written, figures are well-designed.

[2] The idea of synthesizing data for pretraining motion time series is interesting.

[3] 18 real-world datasets are involved for evaluation, multiple activities are included.

**Weaknesses:**

Overall, I think this is an interesting paper and is promising. Therefore, I provide lots of comments. However, it has some obvious potential flaws in the current version. Please see limitations.

**Questions:**

I have some questions regarding the experimental design, baselines, related works, novelty, etc. I put them in the limitation section.

**Limitations:**

[1] The data generation is interesting. I am thinking about what if we replace the generated data with a few real data for model training, how will the performance look like. Or, please see my question in [8], how to justify the importance of synthesized data.

[2] Missing related work. There are important studies of pretraining motion time series missing. Authors can check more carefully, e.g., from sensys, kdd.

[3] For the random rotation method, there are multiple existing approaches. Authors need to justify the difference or add citations. For example, "Device Orientation Independent Human Activity Recognition Model for Patient Monitoring Based on Triaxial Acceleration”, “Data augmentation of wearable sensor data for Parkinson’s disease monitoring using convolutional neural networks”,  and a few other papers from Mobicom and Ubicomp also applied similar methods.

[4] Also, for the rotation method, does Acc and Gyro share the same rotation?

[5] I am wondering is there any data normalization? Given the distribution discrepency among synthesized data and real data, it might hurt the performance. Currently, I only see a normalized matrix in Line 156.

[6] For the data generation module, I like the idea of utilizing physical laws. However, it seems this generation method is not proposed by this manuscript. Therefore, authors might want to justify the technical contribution.

[7] Fig 7 looks weird. Does it mean ImageBind cannot get any samples correct for most activities? Also, how the proposed method and ImageBind infer the activities not seen before should be described in more detail.

[8] The performance gain of the zero-shot setting is impressive. For zero-shot setting, how are baseline models are trained? Are they fine-tuned also with the generated data?

[9] In line 131, it would be great if there is some documents can support the claim.

[10] I guess the authors aim to address the generalization issue of HAR, e.g., device locations, orientations. There are quite a few studies in these area, for domain adaptation  domain generalization. Some of them explore augmentation, some explore pretraining, etc., to address the challenges mentioned in the introduction.

[11] For zero-shot setting, the performance seems too low for real-world usage, e.g., for datasets with only 4 or 6 classes, the acc is smaller than 50%.

[12] Authors mention cross-dataset a few times in the paper, is there some experiments focusing on this setting except the zero-shot experiment?

[13] Another concern I have is the full-shot setting, there are a bunch of SoA models from AAAI, KDD, IMWUT. I am not sure whether the provided baselines are representative. However, it is good to see authors replace some baselines from zero-shot settings.

---

> ### Author Rebuttal · Authors · 2024-08-06
>
> We sincerely thank the reviewer for the valuable feedback and for recognizing our idea, promising results and clear presentation. We have addressed the comments as below.
>
> [1] Importance of synthesized data
>
> We add one experiment by incorporating Capture-24 (one of the largest real data from free-living environments) into pre-training. We randomly sample it to 10% of our synthetic dataset size to mitigate any bias due to its larger size. **Table 2** in the general response PDF shows performance gains mainly in datasets with wrist-worn devices (e.g., WISDM, Wharf) and simpler activities (e.g., UCI-HAR), as Capture-24 was collected with wrist-worn devices and limited activity descriptions. However, overall performance slightly declines due to its single device focus and poor label description quality. This highlights the significance of our synthesized data, which has all joint locations and detailed text descriptions mostly absent in current real-world datasets, facilitating generalization across diverse activities and devices.
>
> [2-3] More related works
>
> We appreciate the additional references (pre-training motion time series [1-3], random rotations [4-6]) and will make sure to include them in the final version. We are also open to including any further related works as suggested by the reviewer.
>
> [4] Rotation
>
> Yes, they share the same rotation. Conversations with leading sensor vendors like STM, Invensense and Bosch confirm that acc and gyro are typically calibrated to have negligible misalignment factors, so we can use the same rotation.
>
> [5] Data normalization
>
> We normalize the data to ensure consistency in unit measurements, e.g., standardizing accelerations to $m/s^2$. We will make this clear in the final version.
>
> [6] Data generation
>
> We acknowledge that there exist simulators that help generate motion time series. However, unlike previous works, we jointly synthesize data across the entire body, and model their correlations via graph encoder for better location generalization. We also align time series with text semantics for activity generalization, presenting the first unified pre-training framework.
>
> [7] Zero-shot generalization
>
> ImageBind pre-trained on Ego4D data with only head-mounted devices cannot generalize to most activities involving different device placements. This highlights the importance of synthesized data covering all joints. UniMTS and ImageBind predict unseen activities by matching time series embeddings with text label candidates, selecting the one with the highest similarity score as shown in Figure 3 of the paper.
>
> [8] Zero-shot setting
>
> Baselines are trained on their original pre-training corpus (e.g., ImageBind, IMU2CLIP on Ego4D) without fine-tuning on synthetic data. We add one experiment by continually training IMU2CLIP on our synthetic dataset as shown in **Table 1** of the general response PDF. IMU2CLIP improves after training on the synthetic data, which again highlights the importance of our synthesized data pre-training. Yet, it still underperforms UniMTS, as our graph encoder better models cross-device spatio-temporal relationship.
>
> [9] Documents
>
> iOS uses inertial force and Android uses accelerating force. We noted such sign discrepancy when collecting data using devices from both OS platforms. This is also confirmed in the Sensor Logger App manual. Rebuttal policy excludes external links, and we will include them in the final version.
>
> [10] Domain adaptation (DA)
>
> Existing DA methods mostly assume identical label names and activity counts between source and target, such as cross-user and cross-dataset DA only for common classes [7-9]. We aim for a more challenging scenario with varying label names across datasets, which existing DA generally cannot address. We will include DA methods as references and make this clear in the final version.
>
> [11] Zero-shot performance
>
> Real-world motion time series classification in a zero-shot manner is extremely challenging, yet we’ve shown a 342.3% improvement over the best baseline. The score also averages \~60% when we retrieve the top-2 activities. Moreover, fine-tuning on 1-shot/10-shot per class further enhances the model to \~60%/\~75% for most datasets.
>
> [12] Cross-dataset experiment
>
> The zero-shot experiment shows how our pipeline generalizes across datasets. It works the same by substituting the synthetic data with others like Capture-24 (see Response [1]). We use synthetic data for its high quality with all-joint coverage and detailed descriptions.
>
> [13] Full-shot baseline
>
> Our compared baselines like GPT4TS (NeurIPS'23), SHARE (CIKM'23), ImageBind (ICCV'23), IMU2CLIP (EMNLP'23) are all SOTA general motion time series classification models. There are more recent papers but they mostly focus on sub-domains such as low-resource [1] or federated learning [10] scenarios instead of general classification.
>
> Reference
>
> [1] Generalizable low-resource activity recognition with diverse and discriminative representation learning, KDD 2023
>
> [2] What makes good contrastive learning on small-scale wearable-based tasks? KDD 2022
>
> [3] Limu-bert: Unleashing the potential of unlabeled data for imu sensing applications, SenSys 2021
>
> [4] Device orientation independent human activity recognition model for patient monitoring based on triaxial acceleration
>
> [5] Data augmentation of wearable sensor data for Parkinson’s disease monitoring using convolutional neural networks
>
> [6] Practically Adopting Human Activity Recognition, MobiCom 2023
>
> [7] SWL-Adapt: An unsupervised domain adaptation model with sample weight learning for cross-user wearable human activity recognition, AAAI 2023
>
> [8] Semantic-discriminative mixup for generalizable sensor-based cross-domain activity recognition, IMWUT 2022
>
> [9] CrossHAR: Generalizing Cross-dataset Human Activity Recognition via Hierarchical Self-Supervised Pretraining, IMWUT 2024
>
> [10] Flames2graph: An interpretable federated multivariate time series classification framework, KDD 2023

---

> > ### Comment · Reviewer_s9ZD · 2024-08-08
> > **Some concerns are addressed**
> >
> > I would like to thank the authors for providing detailed feedback, which addressed some of my concerns.
> >
> > First, the additional experiments are valuable. Results in both Table 1 and Table 2 provide interesting insights about the synthetic data.
> >
> > Second, the additional related works are convincing, I believe these studies will benefit this paper and put this paper in a better position.
> >
> > Third, I hope some explanations could be incorporated in the final version, e.g., the same rotation for gyro and acc.
> >
> > However, I still have one concern about the zero-shot setting in 7 and 8. If methods like ImageBind are only trained with unsupervised pre-training, it will not have the knowledge to connect motion data with corresponding labels. Or, the zero-shot means ImageBind is also fine-tuned with labels (such as walking and running), but zero-shot for recognizing new activities (such as still and biking)?
> >
> > Another concern is the practical impact. Given the low performance of recognizing new activities, I doubt how such performance could generate a practical impact for real-world usage.
> >
> > Again, overall this is a good paper with a good presentation, clear motivation, and high potential for a broad impact. Great work.

---

> > > ### Author Response · Authors · 2024-08-08
> > >
> > > We sincerely thank the reviewer for the insightful comments, and for recognizing that our paper has good presentation, clear motivation, and high potential for a broad impact. We have addressed the concerns as follows.
> > >
> > > [1] References and explanations
> > >
> > > We will make sure to include the additional related works, as well as detailed explanations such as the same rotation for gyro and acc.
> > >
> > > [2] Zero-shot setting
> > >
> > > ImageBind is pre-trained to use image modality as a common bridge to bind multiple modalities together, including motion time series and text. More specifically, the model is pre-trained to align pairwise images and motion time series from the Ego4D daily-life activity dataset, as well as to align large-scale web images and text. By having image modality as the common bridge, the model therefore has the knowledge to connect motion time series with text descriptions. However, as Ego4D only contains motion time series from head-mounted devices, the model struggles to generalize to downstream datasets with different device placements.
> > >
> > > [3] Practical impact
> > >
> > > We would like to note that recognizing new classes in a zero-shot fashion is extremely challenging. Despite the difficulties, UniMTS has outperformed the best existing baseline by an impressive 342.3%. Furthermore, the top-2 retrieval score has increased to approximately 60. In practice, it is often feasible to obtain a few labels for downstream datasets. With just one labeled example per class, the F1 score can improve to about 60, and it further enhances to about 75 with ten labeled examples per class, which significantly outperforms the best baseline by 16.3%. Additionally, even in the full-shot setting, UniMTS continues to outperform the best baseline by 9.2%, making it practically impactful for a broad range of real-world use cases.
> > >
> > > Thank you once again for all your insightful feedback!

---

> > > > ### Comment · Reviewer_s9ZD · 2024-08-11
> > > >
> > > > Dear authors,
> > > >
> > > > Thank you for your timely response!
> > > >
> > > > While I agree with you on some points, some concerns remain.
> > > >
> > > > 1.The pairwise alignment makes sense to me. However, there are still labels needed to enable the model to know, what kind of time series looks like a specific label, say, "walking". Then, in future cases, when the model is given a time series of new labels, say, "running", the model can make the prediction. Therefore, in terms of zero-shot, I guess you mean the model has seen "zero" training samples of data from new categories but still needs some labeled data from at least some labels. Hopefully I have made myself clear.
> > > >
> > > > 2.Your responses make sense to me and I admit the improvement compared to SoAs is significant enough. I still think there is some steps to go to make it practical, at least in the zero-shot setting. But this is totally fine.
> > > >
> > > > Given the comprehensive rebuttal and most of my concerns are addressed, I have raised my score. Hope to see the camera-ready version, great work!

---

> > > > > ### Author Response · Authors · 2024-08-11
> > > > >
> > > > > We sincerely thank the reviewer for the insightful feedback and for supporting our work. We have addressed the concerns as below.
> > > > >
> > > > > [1] We follow the same zero-shot setting as CLIP [1] for both UniMTS and ImageBind. During pre-training, the model learns to connect motion time series with text descriptions on the pre-training dataset. During evaluation, the model predicts new labels on new downstream datasets not seen during training in a zero-shot manner. The term zero-shot refers to the model’s exposure to both new downstream datasets as well as new categories without prior direct training on them.
> > > > >
> > > > > [2] Yes, the zero-shot setting is extremely challenging. Apart from the 342.3% improvement over the best existing baseline, we believe that with the availability of high-quality, large-scale real-world datasets in the future, our framework can be directly applied to model these datasets during pre-training and achieve even more significant improvements.
> > > > >
> > > > > Thank you once again for your support of our work!
> > > > >
> > > > > [1] Radford, A., Kim, J.W., Hallacy, C., Ramesh, A., Goh, G., Agarwal, S., Sastry, G., Askell, A., Mishkin, P., Clark, J. and Krueger, G., Learning transferable visual models from natural language supervision, ICML 2021

---

> > > > > > ### Comment · Reviewer_s9ZD · 2024-08-12
> > > > > >
> > > > > > Thank you for the detailed responses. As you mentioned, the model is only exposed to old datasets and old categories for the training process, which makes sense to me. I am also interested in the new category prediction. Is the model capable of predicting any new categories? Or, you need to pre-define a few new categories (e.g., running and jogging), so the model can link the representation with the pre-defined new categories.

---

> > > > > > > ### Author Response · Authors · 2024-08-12
> > > > > > >
> > > > > > > We sincerely thank the reviewer for the insightful comments. Our text encoder leverages pre-trained weights from CLIP, which inherently understands semantic similarities between words. During pre-training, we didn’t pre-define any categories; we fed the model with extensive text sentence descriptions to align motion time series with CLIP’s semantic space. Some example text descriptions in our pre-training dataset are “the standing person kicks with their left foot before going back to their original stance”, “a man lifts something on his left and places it down on his right”, “a person walks slowly forward then toward the left hand side and stands facing that direction”.
> > > > > > >
> > > > > > > During evaluation, UniMTS leverages such semantic awareness to effectively generalize to new activities. For example, during pre-training the model has encountered “walking” and “running”, during evaluation it can leverage the knowledge that “jogging” represents a movement with intensity level in between “walking” and “running” to successfully generalize and recognize “jogging”. Other examples include generalizing from “ascending stairs” to “walking on inclined treadmill”, and from “waving hands” to “drawing circle counter clockwise”.
> > > > > > >
> > > > > > > We sincerely thank the reviewer once again for the invaluable feedback and support.

---

### Official Review · Reviewer_6pyv · 2024-07-07

**Soundness:** 3
**Presentation:** 4
**Contribution:** 4
**Rating:** 7
**Confidence:** 4

**Summary:**

This paper proposes a unified pre-training framework using simulated data from body skeleton model for motion time series. The authors rightly pointed out the challenges of collecting motion time series at scale due to privacy concerns despite the fact that the motion time series data have great promises in a wide range of applications. Thus, this is an area with few pre-trained models unlike other data modalities.

The simulation framework proposed was used to generate the synthetic data needed for pre-training. The author then utilised the generated motion time series in contrastive learning using a graph neural network that learns a joint embedding for sensor data at different body locations. The extensive evaluations of the pre-trained model demonstrated superior performance for human activity recognition in several settings, including zero-hot, few-shot and full-shot performance.

**Strengths:**

* The manuscript was well-written. I thoroughly enjoyed reading it. I particularly like that the paper clearly points out the methodological gap for motion time series data to the ML audience at NeurIPS.
* The simulation framework is an important contribution to the ubiquitous computing community, for which large-scale datasets are difficult to obtain.
* The use of GNNs to learn a joint embedding for wearable sensors is novel. It is great that the GNN implementation can also account for downstream tasks by using a masking trick when not all sensor modes are available!
* The performance evaluations were extensive in terms of both benchmark datasets and baseline models included.

**Weaknesses:**

1. Despite my general excitement about this paper, this paper has not recognised the body of work already done in large-scale pre-trained models for motion time series. The authors might not be familiar with this area of work because existing pre-trained models generally use Biobank and are published in journals. It would be great if the authors could compare the following work in the context of their results:
    1. Yuan, Hang, et al. "Self-supervised learning for human activity recognition using 700,000 person-days of wearable data." NPJ digital medicine 7.1 (2024): 91.
    2. Spathis, Dimitris, et al. "Self-supervised transfer learning of physiological representations from free-living wearable data." Proceedings of the Conference on Health, Inference, and Learning. 2021.
2. Pre-training from synthetic data is a cost-effective approach. Its performance is likely to plateau at some point as compared to pre-trained using real data. It would be interesting to see a comparison with Yuan et al. 2024 as this model was pre-trained on 700,000 person-days of real data. I also appreciate that adding an experiment might not be realistic, but I do believe that it should be fairly easy to do the fine-tuning.
3. The HumanML3D human selection data only spans over a mere 29 hours. I am very suspicious that any pre-training on just 29 hours of synthetic data will be enough for pre-training as scaling laws go. Even though the current results show good performance, if I am not wrong, the majority of the benchmarks were collected in the lab, which doesn't represent realistic human behaviour in a free-living environment. CAPTURE-24 could be another interesting benchmark as it is one of the largest open-access free-living activity recognition datasets. More importantly, it would be good to get a sense of how the downstream performance scales to your pre-training data so that we can determine whether the benefits of pre-training have plateaued.
4. Also, you probably don't have to add the references below, but I just want you to know that there are several biobanks that contain large-scale wearable sensor data that can be used for pre-training. Your work is still valuable as your can support multi-sensors, whilst most of the existing large-scale datasets only contain wrist-worn devices.
	1. Doherty, Aiden, et al. "Large scale population assessment of physical activity using wrist worn accelerometers: the UK biobank study." PloS one 12.2 (2017): e0169649.
	2. Chen, Yuanyuan, et al. "Device-measured movement behaviours in over 20,000 China Kadoorie Biobank participants." International Journal of Behavioral Nutrition and Physical Activity 20.1 (2023): 138.
	3. Master, Hiral, et al. "Association of step counts over time with the risk of chronic disease in the All of Us Research Program." Nature medicine 28.11 (2022): 2301-2308.

**Questions:**

1. What your input data format during fine-tuning and sampling frequency?
2. Are Model weights and pre-trained model made available?

**Limitations:**

The authors have addressed this adequately.

---

> ### Author Rebuttal · Authors · 2024-08-06
>
> We greatly appreciate the reviewer for all the valuable feedback and for recognizing our technical contributions, novelty of our method, thoroughness of our experiments and the clarity of our paper presentation. We have addressed the comments as below.
>
> [1] Baselines trained on BioBank
>
> We thank the reviewer for mentioning these related works. We will make sure to include them in the final version. We have conducted experiments to compare UniMTS with the most recent work of Yuan et al. 2024 which is pre-trained on the BioBank dataset. We would like to emphasize that
>
> a. The BioBank dataset is limited to signals collected from wrist-worn devices, offering less diversity compared to our synthetic dataset that covers joint locations across the entire body.
>
> b. Yuan et al. 2024 is pre-trained through self-supervised learning and does not involve text semantics aligning, so it cannot be applied for zero-shot recognition as UniMTS does. This highlights the advantages and generalizability of UniMTS.
>
> We compare the few-shot fine-tuning and full-shot performances of UniMTS vs Yuan et al. 2024 respectively in **Figure 1** and **Table 3** of the attached PDF in the general response. UniMTS performs consistently better in both few-shot and full-shot scenarios, demonstrating the effectiveness of our pre-training pipeline.
>
> [2] Add Capture-24 in pre-training
>
> We have conducted experiments by incorporating Capture-24 dataset into pre-training. We randomly sample it to 10% of our synthetic dataset size, in order to mitigate any bias due to its larger size. We report the zero-shot performance of UniMTS after pre-training with Capture-24 in **Table 2** of the attached PDF in the general response.
>
> Notably, performance improvements are primarily observed in downstream datasets that utilize devices attached near the wrist (e.g., WISDM, Wharf) and datasets with simpler activities (e.g., UCI-HAR), which correlates with Capture-24’s collection procedure with wrist-worn devices and limited activity descriptions. However, there is a slight decline in overall performance, attributed mainly to the dataset's focus on a single device location and the poorer quality of label descriptions.
>
> These findings highlight the significance of our synthetic data pre-training, which has high-quality all-joint coverage and detailed, diverse text descriptions mostly absent in current real-world datasets, facilitating generalization across varied activities and devices. We also note a similar trend when increasing the proportion of Capture-24 in our pre-training corpus.
>
> [3] New references
>
> We thank the reviewer for pointing to these related works. We will ensure to add these references in the final version.
>
> [4] Data format during fine-tuning and sampling frequency
>
> The input data are of shape [batch size, sequence length, number of joints, number of feature channels] during fine-tuning. We assign data to specific joint locations according to where the downstream devices are attached, and assign zeros to the remaining joint locations. The sampling frequency is 20 Hz, consistent with the HumanML3D dataset, which is enough for activity recognition tasks.
>
> [5] Release of pre-trained model weights
>
> We will release both pre-trained model weights and all the code upon acceptance of the paper.

---

> ### Comment · Reviewer_6pyv · 2024-08-07
>
> Thank you for all the additional experiments that provided pre-training based on synthetic data compared to biobank pre-training.
> I just have two more comments and questions:
>
> 1. In your Figure 1 comparing BioBankSSL and UniMTS, the few-shot few-tuning performance might be too good to be true, at least in the first instance. How are you doing the train/test split? Currently, the axis is the number of samples from each activity class. If your train/test split is not subject-wise, then the test result is biased.
>
> 2. When you incorporated real data for pre-training (wrist-worn data), your evals on wrist datasets improved despite the evals on other datasets decreased. This is telling me that realistic data is still better than synthetic data if available. I wouldn't expect a model pre-trained on the wrist to work well with other placements, for example. It may be worth discussing the value of realistic and synthetic data in different use cases. Given this observation, I was surprised to find that your BioBankSSL experiments showed that synthetic pre-training on wrist data did better.
>
> Overall, human activity recognition has poor benchmark datasets, often extremely limited in scale (n<50 participants). I wouldn't hold the authors accountable for addressing the benchmark issue for the community. It will take time, but once this paper is accepted and externally validated by other researchers, we can better test the generability of the proposed framework.
>
> You have done a great job addressing most of my concerns, and I have adjusted my score from 6->7.
>
> Obviously, it would be great if you could reply to my new queries above :D

---

> > ### Author Response · Authors · 2024-08-08
> >
> > Dear Reviewer,
> >
> > Thank you so much for your insightful comments and for raising the score! We have answered your questions as follows.
> >
> > [1] Few-shot fine-tuning performance
> >
> > For datasets with a public train/test split (e.g., UCI-HAR), we adopt the available divisions. For the remaining datasets, we split them based on subjects, except for only very few datasets where we didn’t find subject specification (e.g., UT-Complex). Our test window size is 10 seconds, which carries sufficient information for the model to infer the activities and to minimize bias from random noise.
> >
> > UniMTS shows great generalizability in few-shot scenarios due to its pre-training on our synthesized data with diverse device locations, orientations and activity types. In contrast, without such pre-training on diverse device latent factors and text semantics aligning, Yuan et al. 2024 initially performs near-randomly (e.g., achieving an F1 score of 25 for 4-class Opportunity dataset). However, self-supervised pre-training on large-scale BioBank data also provides good initialization for Yuan et al. 2024 to converge faster, so the model is able to quickly improve its performance as more samples per class are given.
> >
> > [2] Discussion on synthetic and real-world data
> >
> > Yes, we believe that the community would greatly benefit from large-scale real-world data with comprehensive joint coverage and detailed text descriptions, if such a dataset becomes available in the future. Should such datasets become available, our framework of graph encoding and contrastive semantic aligning is still directly applicable to model these new datasets, and incorporating these data into our pre-training framework will potentially lead to more significant improvements. However, in the absence of such comprehensive real-world data, currently synthetic data still present higher quality than real-world datasets with limited joint coverage and limited text descriptions. Therefore, we generate synthetic data as the first step to address the challenges, establishing the first pipeline for pre-training motion time series. We believe our approach will bring insights to the community to address this long-standing challenge of data generalization, and we will make sure to discuss these comparisons between synthetic and real-world data in the final version.
> >
> > For the full-shot experiments of BioBank SSL, we observe that datasets with wrist-worn devices, such as WISDM and Wharf, also show slightly better performance compared to UniMTS, which is consistent with the observation that incorporating Capture-24 slightly improves performance on these wrist-worn datasets. For few-shot experiments, given that BioBank SSL is not pre-trained to align with text semantics, it would be hard for it to generalize to new activities given few shots, even for datasets with wrist-worn devices. Therefore, UniMTS consistently outperforms BioBank SSL in the few-shot scenarios, demonstrating the significance of our pre-training pipeline.
> >
> > Thank you once again for your invaluable feedback!

---

> ### Comment · Reviewer_6pyv · 2024-08-08
>
> Thank you very much for the helpful feedback.
>
> Your evals make sense. I fear that the issue with existing benchmark is that it is very easy to overfit on tiny test set (data with a few subjects) with a large model. So I wouldn't necessarily trust the test results even though they conform with existing literature for consistency sake.
>
> This work is of high quality and has high impact to the field of ubiquitous computing and moderate impact on several other fields including pre-training using synthetic data and multi-modal learning.
>
> I thoroughly enjoyed reading this paper. Looking forward to seeing the final version this manuscript and the open release of the model!

---

> > ### Author Response · Authors · 2024-08-08
> >
> > Thank you so much for your insightful comments and for recognizing that our work is of high quality and high impact!
> >
> > We have tried our best to evaluate the model on a diverse set of real-world datasets, and to reduce bias by splitting train/test data based on subjects. We believe that future availability of larger-scale benchmarks will further validate the robustness of our model. We greatly appreciate your interest in our work and are committed to releasing the model upon acceptance.
> >
> > Thank you once again for all your invaluable feedback!

---

### Official Review · Reviewer_G59f · 2024-07-08

**Soundness:** 4
**Presentation:** 4
**Contribution:** 3
**Rating:** 7
**Confidence:** 3

**Summary:**

In the context of time series tasks, the paper addresses motion time series data collected from mobile and wearable devices such as smartphones and smartwatches. However, training and testing on the same dataset leads to poor generalizability of the models. The authors propose UniMTS, a unified pre-training procedure for motion time series that generalizes across diverse device latent factors and activities.

**Strengths:**

- The paper is well written and easily understandable.
- The soundness of the technical claims are good and the experiments support the proposed method.
- The problem setup is well motivated. The addressed problem exists in many application areas and research fields.
- The paper proposed a large experiments on many different datasets showing a higher performance than state-of-the-art methods.

**Weaknesses:**

- After reading the Abstract, the actual contribution of the paper is unclear. The Abstract describes several minor contributions, but what is the focus of the paper?
- It is unclear which large language model is used (should be mentioned earlier).
- Does the proposed method actually fit to NIPS? The contributions are limited/unclear.

**Questions:**

- Why cannot domain adaptation be utilized to address the shift in data or continual learning to adapt to new activities, etc.?

**Limitations:**

- The limitation is that the main contribution of the paper is utilizing LLMs to improve the generalizability of time series models.

---

> ### Author Rebuttal · Authors · 2024-08-06
>
> We express our gratitude for the reviewer’s constructive feedback, and for recognizing the soundness of our technical claims, the scale and performance of our experiments, as well as the clarity of our paper’s presentation. We have addressed each of the concerns as below.
>
> [1] Contribution of the paper
>
> a. We propose the **first unified pre-training framework** for motion time series. While large-scale pre-training is common in vision and audio domains, it is challenging to develop similar pre-trained models given the diversity of motion time series data due to variations in device locations, device mounting orientations, and human activity types across different datasets. Existing works mostly train and test their models on a single type of datasets with limited generalizability. Our proposed UniMTS framework **for the first time** addresses all these challenges and generalizes to diverse downstream datasets.
>
> b. **UniMTS generalizes across datasets with various device locations**. Real-world motion time series datasets typically focus on different device locations, and no single real-world dataset provides comprehensive location coverage. To address this gap, we propose to synthesize motion time series from motion skeletons with all-joint coverage. By utilizing a graph encoder, we effectively model the spatio-temporal correlations across these joints, enhancing the generalizability of UniMTS across diverse device locations.
>
> c. **UniMTS also generalizes across device orientations**. The device mounting orientations could significantly affect the downstream performance. To ensure the model remains robust regardless of how the devices are oriented, we propose to use rotation-invariant augmentation techniques during pre-training.
>
> d. **Additionally, UniMTS generalizes across different activity types**. We align motion time series with text semantics through contrastive learning, enabling classification beyond predefined class labels and supporting arbitrary activity types. To improve the diversity and generalization of activity type representations, we further use text augmentations from LLMs.
>
> **We would like to highlight that utilizing LLMs is *not* our main contribution**. Our main contribution is the novel pre-training framework for motion time series, which represents a significant advancement unseen in existing literature. LLMs are just one of the techniques we use to augment text descriptions for enhanced pre-training activity diversity. In addition to this technique, we have proposed various methods, as outlined above, to improve generalization across multiple aspects.
>
> [2] Which LLM is used
>
> We use GPT-3.5 (“gpt-3.5-turbo”) as discussed in Section 4.1 in the paper. We will make sure to mention this in earlier sections in the final version.
>
> [3] Paper scope
>
> Our work is of significant relevance to various key topics that have a wide audience in NeurIPS, including time series analysis, machine learning for healthcare, pre-training, physics-based simulation, and motion modeling. One of the key challenges in these communities is to improve model’s generalizability across diverse datasets. Our contribution falls under this area by designing the first unified pre-training procedure for motion time series, which successfully generalizes to various device locations, device orientations and activities. We believe our insights are valuable to the community and align well with the scope of NeurIPS.
>
> [4] Domain adaptation and continual learning methods
>
> Existing domain adaptation methods mostly assume that source and target datasets share the same label names and have the same number of activities, such as cross-user domain adaptation and cross-dataset domain adaptation **only for those common classes** [1-3]. We aim for a more generic yet challenging generalization scenario where pre-training and downstream datasets share different label names. Therefore, existing domain adaptation methods generally cannot be applied to address the activity generalization challenge that our work seeks to overcome. Moreover, continual learning [4] typically involves training on the new activities with the risk of catastrophic forgetting. In contrast, our model is able to generalize to new activities in a zero-shot fashion. We will include domain adaptation and continual learning methods as references and make this clear in the final version.
>
> Reference
>
> [1] Hu, R., Chen, L., Miao, S., & Tang, X, SWL-Adapt: An unsupervised domain adaptation model with sample weight learning for cross-user wearable human activity recognition, AAAI 2023.
>
> [2] Lu, W., Wang, J., Chen, Y., Pan, S. J., Hu, C., & Qin, X, Semantic-discriminative mixup for generalizable sensor-based cross-domain activity recognition, IMWUT 2022.
>
> [3] Hong, Z., Li, Z., Zhong, S., Lyu, W., Wang, H., Ding, Y., He, T. and Zhang, D., CrossHAR: Generalizing Cross-dataset Human Activity Recognition via Hierarchical Self-Supervised Pretraining, IMWUT 2024.
>
> [4] Jha, S., Schiemer, M., Zambonelli, F. and Ye, J., Continual learning in sensor-based human activity recognition: An empirical benchmark analysis, Information Sciences, 2021.

---

> > ### Comment · Reviewer_G59f · 2024-08-11
> >
> > Thank you for your comprehensive response and the further clarifications provided. The paper's contribution is now well understood. The primary contribution lies in the innovative pre-training framework, which enables generalization across datasets with varying device locations, orientations, and different activity types. The experiments presented substantiate this contribution. I also appreciate the additional references provided.
> >
> > All my concerns have been addressed, and I have increased my rating to acceptance, recognizing the potential for broad impact.

---

> > > ### Author Response · Authors · 2024-08-11
> > >
> > > We sincerely thank the reviewer for all the valuable feedback and for recognizing the potential broad impact of our work. We greatly appreciate your support!

---

### Official Review · Reviewer_7idQ · 2024-07-12

**Soundness:** 3
**Presentation:** 3
**Contribution:** 3
**Rating:** 7
**Confidence:** 5

**Summary:**

This paper introduces a unified pre-training procedure for motion time series that generalizes across diverse device latent factors and activities. A contrastive learning framework that aligns motion time series with text descriptions enriched by large language models and the spatio-temporal graph networks are utilized to capture the relationships across joints for generalization across different device locations. Experimental results show the superior performance of the proposed method that selected baselines.

**Strengths:**

1. The paper is well-written and easy to follow.
2. The paper propose a unified pre-training framework for motion time series based on contrastive learning. Specifically, the proposed framework introduce LLM to incorporate the knowledge of text descriptions into time series modeling.
3. Extensive experiments demonstrate the effectiveness of the proposed method.

**Weaknesses:**

1. It would be better to provide more descriptions regarding the usage of LLM.
2. It would be better to discuss the space and time complexities of the proposed framework.
3. As efficiency is important in real-world applications. It would suggest conducting experiments to compare the proposed method's training time (including pre-training time) with baselines.

**Questions:**

1. What is the motivation for the usage of LLM?
2. Can the proposed method improve the efficiency of existing methods (e.g., baselines)?
3. How do you obtain the text descriptions?

**Limitations:**

Please see the weaknesses and questions.

---

> ### Author Rebuttal · Authors · 2024-08-06
>
> We sincerely thank the reviewer for the valuable comments and for recognizing the effectiveness and clarity of our work. We have addressed each comment as below.
>
> [1] How to obtain text descriptions
>
> We use the original text descriptions in the motion skeleton dataset HumanML3D. As detailed in Appendix A.1 of our paper, there is an average of 3 textual descriptions paired with each motion skeleton sequence in the HumanML3D dataset, resulting in a total of 44,970 textual descriptions with a vocabulary size of 5,371. Some example text descriptions are “the standing person kicks with their left foot before going back to their original stance”, “a man lifts something on his left and places it down on his right”, “a person walks slowly forward then toward the left hand side and stands facing that direction”.
>
> [2] Usage of LLM
>
> We apply LLM to generate paraphrases of original text descriptions in the HumanML3D dataset to further increase the diversity. This allows UniMTS to better learn the text semantics and generalize to varied label descriptions in the downstream datasets. For instance, consider the initial description of a skeleton motion: “a person falls to the ground in a sitting motion and then pops back up in a standing position”. From this, the LLM generates three distinct paraphrases: “a person drops into a seated position before quickly rising to their feet”, “a man crouches down and quickly jumps up”, and “someone squats low and then springs up”. These variations substantially broaden the range of text descriptions, helping the model in generalizing across various label names in the downstream datasets.
>
> [3] Space and time complexities compared with baselines
>
> a. For space complexity, our graph encoder contains only 4.94M parameters, which is significantly smaller compared with the 18.69M used in the IMU encoder of the best existing baseline ImageBind.
>
> b. For time complexity, our pre-training takes a one-time 32 hours. Our fine-tuning is also efficient. On one example dataset of UCI-HAR, full-shot fine-tuning of UniMTS takes ~1.3 minutes to converge while it takes ~9.8 minutes for ImageBind to converge. Moreover, we have run a power estimate assuming 0.1Hz cadence (i.e., 10-second window size), and it takes ~22.64 mW to run the whole graph model on an eNPU (embedded Neural Processing Unit), which is much smaller than ImageBind IMU encoder’s power consumption of ~702 mW. Therefore, our model is efficient for real-world applications and suitable to be deployed on edge devices.
>
> [4] Our proposed method can improve the efficiency of existing methods
>
> a. As we discussed in Response [3], UniMTS is more efficient than baselines in terms of both time and space complexities, and converges faster during fine-tuning.
>
> b. Our synthetic data pre-training also provides an effective initialization which further improves the fine-tuning efficiency. For example, fine-tuning on downstream datasets takes on average 10 epochs to converge. Moreover, such benefits are model-agnostic and we can also improve the fine-tuning efficiency of baseline models when they are pre-trained on our synthetic data.

---

> > ### Comment · Reviewer_7idQ · 2024-08-09
> >
> > Thanks to the authors for answering my questions. I am satisfied with the responses except for the Space and time complexities compared with the baselines. I mean the theoretical space and time complexities analysis. Thanks.

---

> ### Author Response · Authors · 2024-08-10
>
> We sincerely thank the reviewer for the insightful feedback. We compare the theoretical space and time complexities with the best-performing baseline ImageBind as follows. Since UniMTS and ImageBind share similar text encoders, our analysis mostly focuses on their respective IMU encoders. Let $B$, $L$, and $F$ represent the batch size, sequence length, and feature dimension, respectively. Additionally, let $V$ and $V'$ denote the total number of joints (with $V=22$ for the HumanML3D dataset) and the number of joints with attached devices in downstream datasets ($V' \leq V$). We note that the skeleton graph is sparse and the number of edges is $E=V-1$.
>
> For UniMTS, the time complexity is $\mathcal{O}(BVLF^2 + BVLF)$ and the space complexity is $\mathcal{O}(BVLF + F^2 + V)$. For ImageBind, the time complexity is $\mathcal{O}(BV'LF^2 + BV' L^2F)$ and the space complexity is $\mathcal{O}(BV'LF + F^2 + BV'L^2)$. We observe that UniMTS does not contain the $L^2$ terms present in ImageBind (i.e., complexity grows linearly rather than quadratically with respect to sequence length), making it more efficient in terms of both time and space complexities.

---

> > ### Comment · Reviewer_7idQ · 2024-08-11
> >
> > Thank you for your timely response. Combining your previous actual space and time complexity, your analyses are comprehensive. I have raised my score. Nice work!

---

> > > ### Author Response · Authors · 2024-08-11
> > >
> > > We sincerely thank the reviewer for all the insightful feedback and we greatly appreciate your support!

---

### Official Review · Reviewer_jvaP · 2024-07-12

**Soundness:** 3
**Presentation:** 3
**Contribution:** 3
**Rating:** 6
**Confidence:** 3

**Summary:**

This paper presents a novel pretraining method for IMU time series (i.e., acceleration + angular velocity). The authors make use of human skeleton sequences and simulate IMU measurements by calculating the acceleration and angular speed at each joint. These simulated data are then used to pre-train a spatial-temporal graph encoder. Since each skeleton is paired with texts, the graph encoder is trained with contrastive learning against CLIP text features. Augmentations on IMU data and texts are performed to boost the performance.

**Strengths:**

1. The idea of generating IMU measurements from skeleton sequences is very interesting. On one hand, it brings a huge amount of plausible motion time series for pre-training. On the other hand, it could draw connections between IMU and other modalities.

2. The authors conduct extensive experiments and the proposed method consistently outperforms other baselines in both zero-shot, few-shot, and full-shot settings. The results look promising to me.

**Weaknesses:**

1. I have some concerns about the quality of the generated IMU data as it is constrained to 20Hz due to the original skeleton sequences. This sampling frequency might be enough for action recognition but it is unknown whether it can be used for other tasks such as inertial navigation. Besides, some sensor-level distortion such as IMU drift or some spike noises cannot be simulated using the proposed method.

2. What is the model size of the graph encoder? Since the model is trained on IMUs from many joints (>20?), the model might be huge when fine-tuned on a dataset that has fewer IMU sensors (e.g., 2 sensors), which could make the model inefficient for real-world applications.

**Questions:**

Please refer to weakness

**Limitations:**

The authors have adequately addressed the limitations

---

> ### Author Rebuttal · Authors · 2024-08-06
>
> We sincerely thank the reviewer for the valuable feedback, and for recognizing the novelty of our approach and the promising results from our experiments. We have addressed each of the comments as below.
>
> [1] Quality of generated IMU data
>
> UniMTS is aimed for action recognition, so the sampling frequency of 20Hz is sufficient. We leave other tasks such as inertial navigation as future work as we discussed in the “Limitation and Future Work” section in the paper. Our current simulation process has incorporated IMU drift as offset vectors. We will make this clear in the final version. Spike noises are relatively uncommon to occur over a long period of observation window (e.g., a few seconds), especially considering that modern wearable devices have built–in low pass filters, and thus have minimal impact on downstream classification. Therefore, we have chosen to incorporate the more common Gaussian noise during simulation, as specified in Equation 4 in the paper. We will leave exploring other types of noises as future work.
>
> [2] Efficiency of graph encoder
>
> Our graph encoder contains only 4.94M parameters, which is significantly smaller compared with the 18.69M used in the IMU encoder of the best existing baseline ImageBind. On one example dataset of UCI-HAR, full-shot fine-tuning of UniMTS takes ~1.3 minutes to converge while it takes ~9.8 minutes for ImageBind to converge. Moreover, we have run a power estimate assuming 0.1Hz cadence (i.e., 10-second window size), and it takes ~22.64 mW to run the whole graph model on an eNPU (embedded Neural Processing Unit), which is much smaller than ImageBind IMU encoder’s power consumption of ~702 mW. Therefore, our model is efficient for real-world applications and suitable to be deployed on edge devices.

---

> > ### Comment · Reviewer_jvaP · 2024-08-10
> > **Response to the authors**
> >
> > I would like to thank the authors for their responses. Most of my concerns have been resolved. It is a good work with potential broad impact. Look forward to the release of dataset and pre-trained models.

---

> > > ### Author Response · Authors · 2024-08-10
> > >
> > > We sincerely thank the reviewer for all the insightful feedback, and for the recognition of the quality and potential broad impact of our work. We are committed to releasing the datasets and pre-trained models upon acceptance of the paper. Thank you once again for your valuable feedback and support.

---

### Official Review · Reviewer_4CUw · 2024-07-14

**Soundness:** 3
**Presentation:** 3
**Contribution:** 3
**Rating:** 7
**Confidence:** 4

**Summary:**

The paper introduces a new pre-training approach designed specifically for motion time series data from mobile and wearable devices. The proposed method, UniMTS, employs a contrastive learning framework to align motion time series with text descriptions, enhanced by large language models. This approach addresses the challenges of device placement variation, orientation differences, and activity variability. The model is evaluated on 18 benchmark datasets and demonstrates significant improvements in zero-shot, few-shot, and full-shot settings.

**Strengths:**

Originality:
The paper presents a highly original approach by combining contrastive learning with synthetic data generation and graph networks to pre-train on motion time series data. This is a novel application of existing techniques to a new domain, addressing the unique challenges posed by motion data.

Quality:
The methodology is well-developed and the experiments are thorough. The authors provide a detailed description of their approach, including the use of synthetic data and graph networks. The evaluation on a wide range of datasets strengthens the validity of their findings.

Clarity:
The paper is clearly written and well-organized. Each section logically follows from the previous one, making the complex methodology easy to understand. The appropriate use of figures and tables helps illustrate key points and results effectively.

Significance:
UniMTS addresses a critical gap in the field of motion time series analysis. By improving generalization across different device placements, orientations, and activities, this method has the potential to significantly impact applications in healthcare, sports, and IoT. The impressive performance improvements demonstrated in the experiments underscore the method's potential.

**Weaknesses:**

Evaluation Comparisons:
The paper could be strengthened by comparing the proposed method with more recent benchmarks such as Ego4D and Exo. This would provide a clearer picture of how UniMTS stands relative to the current state-of-the-art methods in the field.

Generalizability of Results:
While the method shows strong results across multiple datasets, there is limited discussion on the potential limitations or specific conditions where the model might underperform. A more detailed analysis of the generalizability and potential edge cases would be beneficial.

**Questions:**

Can you provide a comparison with the Ego4D and Exo benchmarks?

---

> ### Author Rebuttal · Authors · 2024-08-06
>
> We sincerely thank the reviewer for the insightful feedback, and for recognizing the novelty, experimental rigor, and clarity of our presentation. We have addressed the comments as below.
>
> [1] Ego4D and Ego-Exo4D benchmark
>
> a. We acknowledge that these are important action recognition benchmarks; however, we would like to highlight that these two benchmarks are mainly aimed for multimodal action recognition by incorporating video and audio modalities [1,2]. The motion time series in these datasets are collected from head-mounted devices, and recognizing activities based purely on head movement is a very challenging task. Therefore, only very few papers target this task, such as IMU2CLIP. For example, IMU2CLIP reports an F1 score of only 31.89 for a 4-class action recognition task.
>
> b. We choose Ego4D as an example benchmark for comparison with IMU2CLIP. It’s important to note that models on the Ego4D benchmark are already pre-trained on the Ego4D dataset, but our UniMTS has never seen the Ego4D dataset during pre-training. Therefore, for a fair comparison, we split the Ego4D dataset into a 80% training set and a 20% test set, continue pre-training of UniMTS on the training set, and compare with IMU2CLIP pre-trained on the same training set. We report both accuracy and F1 in the following table. As the original Ego4D dataset only contains text descriptions without label IDs, for the test set, we derive class labels by extracting stem verbs from these original text descriptions, and choose top 10, top 20, top 50 activities for evaluation. During evaluation, for both UniMTS and IMU2CLIP, we compute the similarity of signal embeddings and embeddings of each label candidate. The activity with the highest similarity score is predicted as the label.
>
> Setting|Top-10 Acc|Top-10 F1|Top-20 Acc|Top-20 F1|Top-50 Acc|Top-50 F1
> ---------|---------------|--------------|---------------|--------------|--------------|--------------
> IMU2CLIP|25.2|24.6|16.3|15.5|10.8|8.5
> UniMTS|**29.1**|**26.8**|**18.9**|**15.7**|**12.0**|**9.4**
>
> We observe that UniMTS also consistently outperforms the current state-of-the-art method on the Ego4D benchmark, demonstrating the effectiveness and generalizability of our pre-training framework.
>
> c. We will explore combining visual features with inertial signals for action recognition as future work. We believe that inertial signals will help the visual features to learn implicit physical constraints, potentially leading to better generalization. We will also include these benchmarks as references in the final version.
>
> [2] Generalizability and potential edge cases
>
> As we discussed in the “Limitation and Future Work” section in the paper, the simulated signals are for body joints, but real signals might be collected near – rather than directly on – the body joints. For example, sensors on smartwatches collect data near the wrist, not on the wrist joint itself. We plan to incorporate random offset vectors to better simulate real-world signal variations near joints. Other potential edge cases include missing data and multi-device synchronization for real-time motion time series classification, which we plan to address by simulating missing data across time and devices during pre-training.
>
> Reference
>
> [1] Tan, S., Nagarajan, T. and Grauman, K., Egodistill: Egocentric head motion distillation for efficient video understanding, NeurIPS 2023.
>
> [2] Liu, M., Ma, L., Somasundaram, K., Li, Y., Grauman, K., Rehg, J.M. and Li, C., Egocentric activity recognition and localization on a 3d map, ECCV 2022.

---

> > ### Comment · Reviewer_4CUw · 2024-08-12
> >
> > I want to thank the authors for their thorough response and new experiments.

---

> > > ### Author Response · Authors · 2024-08-12
> > >
> > > We sincerely thank the reviewer for all the insightful feedback and for supporting our work!

---

### Author Rebuttal · Authors · 2024-08-07

We sincerely thank all the reviewers for their valuable comments and feedback. We appreciate that nearly all the reviewers recognized the strengths of our work as follows:

a. **Novelty**: high original approach (4CUw), novel method (jvaP, 6pyv).

b. **Importance**: address a critical gap, significant impact (4CUw), well motivated (G59f), important contribution to the community (6pyv).

c. **Promising results supported by extensive experiments**: thorough experiments, impressive performance improvements (4CUw), extensive experiments, promising results (jvaP), superior performance (7idQ, 6pyv, s9ZD), sound technical claims supported by large experiments showing higher performance than SOTA (G59f).

d. **Clarity**: clearly written and well-organized (4CUw), thoroughly enjoyed reading (6pyv), well-written (7idQ, G59f, 6pyv, s9ZD).

We have addressed all the comments from the reviewers, and the major results are as below.

a. We observe better performance on the Ego4D benchmark (4CUw).

b. We show that UniMTS is efficient in terms of both space and time complexities (jvaP, 7idQ).

c. We clarify that our main contribution is designing the first unified pre-training framework for motion time series that generalizes to various downstream device latent factors and activities (G59f).

d. We show better performance compared with pre-trained models on BioBank (6pyv).

e. Our synthetic data generation is critical and model-agnostic. It can also improve the training efficiency for baselines (s9ZD).

f. We show the importance and diversity of our synthesized data by comparing it with pre-training that incorporates real-world datasets with limited joint coverage and text semantics (Capture-24) (6pyv, s9ZD).

---

### Decision · Program_Chairs · 2024-09-25

**Decision:**

Accept (poster)

**Comment:**

The paper proposes a pre-training strategy, specifically designed for motion time series from mobile sensors. The contributions lie in a contrastive learning framework and the alignment of time series with textual descriptions from LLMs. IMU measurements are simulated for each joint and used to pre-train a graph encoder. Doing so, the approach addresses common technical challenges such as variations in the device placements, difference in sensor orientations, and variable activity. On a large benchmark study of 18 datasets the authors show significant improvements over the state of the art in key metrics and in settings such as zero-sho, few-shot and full-shot. As a results, the approach also allows to use a pre-trained model that generalizes well to slight variations in the setup (e.g. across orientations, placements, ...). The reviewers mainly raised questions about the clarity of technical details which have all been resolved during the rebuttal period.

Decision:
After an intensive discussion all reviewers voted for clear acceptance. I therefore recommend to accept the paper and encourage the authors to use the feedback provided to improve the paper for its final version.